# Exploring the Latent Space of Autoencoders with Interventional Assays

Felix Leeb*†        Stefan Bauer †        Michel Besserve †        Bernhard Schölkopf †

## Abstract

Autoencoders exhibit impressive abilities to embed the data manifold into a low-dimensional latent space, making them a staple of representation learning methods. However, without explicit supervision, which is often unavailable, the representation is usually uninterpretable, making analysis and principled progress challenging. We propose a framework, called *latent responses*, which exploits the locally contractive behavior exhibited by variational autoencoders to explore the learned manifold. More specifically, we develop tools to probe the representation using interventions in the latent space to quantify the relationships between latent variables. We extend the notion of disentanglement to take the learned generative process into account and consequently avoid the limitations of existing metrics that may rely on spurious correlations. Our analyses underscore the importance of studying the causal structure of the representation to improve performance on downstream tasks such as generation, interpolation, and inference of the factors of variation.

## 1 Introduction

Autoencoders (AEs) [1, 2] and its modern variants like the widely used variational autoencoders (VAEs) [3], are a powerful paradigm for self-supervised representation learning for generative modeling [4], compression [5], anomaly detection [6] or natural language processing [7]. Since autoencoders can learn low dimensional representations without requiring labeled data, they are particularly useful for computer vision tasks where samples can be very high dimensional making processing, transmitting, and search prohibitively expensive. Here VAEs have shown impressive results, often achieving state-of-the-art results compared to other paradigms [8, 9, 10, 11].

The striking performance coupled with the relatively flexible approach has prompted an explosion of variants to learn a representation with some structure that is particularly conducive to a given task [12, 13, 14]. In addition to a meaningful lower-dimensional representations of our world [15], the focus may be to improve generalization [16, 17, 18], increase interpretability by disentangling the underlying mechanisms [19, 20, 21, 22], or even to enable causal reasoning [23, 24].

In designing ever more intricate training objectives to learn more specialized structures as part of complicated model pipelines, it becomes increasingly important to gain a better understanding of what the representation actually looks like to more quickly identify and resolve any weaknesses of a proposed method. Here the manifold learning community provides a principled formulation to analyze and control the geometry of the data manifold learned by the representation [25, 26, 27, 28, 29, 30, 31, 32, 33]. Developing tools to better understand the structure of the representation is not only useful as a diagnostic to identify avenues for improving modelling and sampling [34, 35], but it also has crucial importance for fairness [36, 37] and safety [18, 38, 39].

Our focus here is on taking advantage of common properties of autoencoders to gain a deeper understanding of the structure of the representation. We summarize our contributions as follows:

---

*Email: fleeb@tuebingen.mpg.de
†Max Planck Institute for Intelligent Systems, Tübingen, Germany

36th Conference on Neural Information Processing Systems (NeurIPS 2022).

- We propose a framework, called *latent responses*, which exploits the locally contractive behavior of autoencoders to distinguish the informative components from the noise in the latent space and to identify the relationships between latent variables.
- We develop tools to analyze how the data manifold is embedded in the latent space by estimating the extrinsic curvature which also enables semantically meaningful interpolations.
- Where true labels are available, we use *conditioned latent responses* to assess how each true factor of variation is encoded in the representation and introduce the *Causal Disentanglement Score* to quantify how disentangled the learned generative process is.

We release our code at `https://github.com/felixludos/latent-responses`.

## 2 Background

Representation learning begins with a set of $N$ observation samples $x \in \mathcal{X} \subseteq \mathbb{R}^D$ which originate from some unknown stochastic generative process with distribution $x \sim p(X)$ and support $\mathcal{X}$. This data manifold $\mathcal{X}$ is embedded into a low-dimensional ($d << D$) latent space $\mathbb{R}^d$, and is modelled by the support $\mathcal{Z}$ of an encoder $f : \mathcal{X} \mapsto \mathcal{Z}$ with the decoder learning the inverse mapping $g : \mathcal{Z} \mapsto \hat{\mathcal{X}}$ where after training $\hat{\mathcal{X}} \approx \mathcal{X}$.

**Variational Autoencoders** (VAEs) [3, 40] are a framework for optimizing a latent variable model $p(X) \approx \int_Z p(X \mid Z; \theta)p(Z)\mathrm{d}Z$ with parameters $\theta$, typically with a fixed prior $p(Z) = \mathcal{N}(Z; 0, I)$, using amortized stochastic variational inference. A variational distribution $q(Z \mid X; \phi)$ with parameters $\phi$ approximates the intractable posterior $p(Z \mid X)$. The encoder and decoder are parameterized such that $q(Z \mid X = x; \phi) = \mathcal{N}(Z; f^\phi(x), \sigma^\phi(x))$, and $\mathbb{E}\left[p(X \mid Z = z; \theta)\right] = g^\theta(z)$ where $f^\phi(x)$ and $\sigma^\phi(x)$, and $g^\theta(z)$ are neural networks which are jointly optimized using the reparameterization trick [3] to maximize the ELBO (Evidence Lower BOund) which is a lower bound to the log likelihood:

$$\log p(X; \theta) \geq \mathbb{E}_{q(Z \mid X; \phi)}\left[\log p(X \mid Z; \theta)\right] - D_{\mathrm{KL}}\left(q(Z \mid X; \phi) \| p(Z)\right) = \mathcal{L}_{\theta, \phi}^{ELBO}(X). \quad (1)$$

Note that in practice, $p(X)$ is unknown, and we only have access to samples $\{x^{(i)}\}_{i=1..N}$, so $p(X)$ is approximated by $\pi(X = x) = \frac{1}{N}\sum_{i=1}^{N}\delta(x - x^{(i)})$. The first term in the objective corresponds to a reconstruction loss, while the second can be interpreted as a regularization term to encourage the posterior to match the prior.

### 2.1 Related Work

**Representation geometry** A closely related approach is manifold learning which aims to exploit the geometry of the data manifold, usually by regularizing the geometry of the representation by estimating the intrinsic curvature of the data manifold [26, 27, 28, 29, 30], or by improving sampling and interpolation using the the Riemmanian metric [31, 32, 25, 33]. In comparison, our response maps estimate the extrinsic curvature, which focuses on the specific embedding, rather than being an intrinsic property of the dataset (see appendix A.4).

**Interpretability and Disentanglement** Another general approach is to gain a better understanding of the representation by making it more interpretable, by, for example, disentangling the true factors of variation [19, 22, 41, 42, 43, 44]. While our analysis is more similar to approaches that improve the representation by focusing on the structure of the representation, for example by learning extra post-hoc models in the latent space [45, 46, 47], we develop a metric to evaluate how disentangled the learned generator is, rather than just the encoder.

**Autoencoder Consistency** VAEs have been investigated from an information theoretic viewpoint [48, 49] and with respect to training problems like posterior collapse [50] or the holes problem [51] to better understand common failure modes. Similarly, mismatches between the encoder and decoder [52, 53], have spurred research into increasing the self-consistency of autoencoders [54, 55, 56, 57, 58]. Our latent response framework relies on a very similar approach and formulation, but crucially, thus far these methods focus on regularizing the training of the representation to impose certain desired structure, while we focus on analyzing the structure that is learned rather than modifying the training objective, making our tools directly applicable to virtually all VAEs.

# 3 Latent Responses

On a high level, to explore the structure of how the data manifold is embedded in the latent space, it is necessary to separate the semantic information from the noise in the latent space. So our goal is to decompose the latent variables $Z$ into an endogenous $S$ and exogenous $U$ component, which, for simplicity, we choose to relate to one another as shown in equation 2.

$$Z = S + U \tag{2}$$

Conceptually, $S$ should capture the semantic information necessary to reconstruct the sample $X$, while $U$ is a local noise model in the latent space for a given observation $X$ which does not meaningfully affect the semantics. In the context of VAEs with a Gaussian prior, we propose equations 3 and 4, which recover the familiar posterior for VAEs $q(Z \mid X = x; \phi) = q(S \mid X; \phi)q(U \mid X; \phi) = \mathcal{N}(f^\phi(x), \sigma^\phi(x))$.

$$q(S = s \mid X = x; \phi) := \mathcal{N}(S = s; f^\phi(x), 0) = \delta(f^\phi(x) - s) \tag{3}$$

$$q(U = u \mid X = x; \phi) := \mathcal{N}(U = u; 0, \sigma^\phi(x)) \tag{4}$$

One implication of separating the deterministic and stochastic part of the encoder is a new perspective on the training signal for the decoder from the VAE objective. Substituting our definitions for $S$ and $U$ into the reconstruction loss term (shown in equation 5) reveals the decoder is trained to map all samples from the posterior to match the same observation sample $x^{(i)}$. As a result, the decoder learns to filter out any exogenous component $u$ from $z$ around $s$, making the latent space locally contractive around $S$ to the extent of $U$. Although we observe this as a byproduct of the VAE objective, this contractive behavior is even observed in unregularized autoencoders to some extent [59] suggesting this may be a more fundamental feature of the inductive biases in deep autoencoders.

$$\mathbb{E}_{u \sim q(U \mid X = x^{(i)}; \phi)} \left[ \log p(X = x^{(i)} \mid s^{(i)} + u; \theta) \right] \tag{5}$$

where $s^{(i)} = f^\phi(x^{(i)})$ is the (deterministic) latent code corresponding to the observation $x^{(i)}$.

Starting from some latent sample $z \sim p(Z)$, we would like to separate the constituent exogenous $u$ and endogenous $s$ components. If we had the matching $x$ such that $z \sim q(Z \mid X = x; \phi)$, then the separation would be trivial since, by definition $s = f^\phi(x)$ and $u = z - s$. However, since the VAE is optimized to reconstruct observations from the latent space, we approximate the missing $x$ with $\hat{x} = g^\theta(z)$. Subsequently, we encode the generated sample $\hat{x}$ to infer an approximation of $s$, $\hat{s} = f^\phi(\hat{x})$. Now by expanding $g^\theta$ around $z$ and $f^\phi$ around $x$ to the first order, we are left with three terms shown in equation 6 (neglecting higher order terms, full derivation in appendix A.2).

$$\hat{s} = \underbrace{s}_{\textcircled{1}} + \underbrace{\mathbf{J}_{f^\phi}(x)(g^\theta(s) - x)}_{\textcircled{2}} + \underbrace{\mathbf{J}_{f^\phi}(x)\mathbf{J}_{g^\theta}(s)u}_{\textcircled{3}} + \mathcal{O}\left[\epsilon^2\right] + \mathcal{O}\left[u^2\right] \tag{6}$$

where $\epsilon = \hat{x} - x$ is the reconstruction error, $\mathbf{J}_{f^\phi}(x)$ is the jacobian of $f^\phi$ evaluated at $x$ and $\mathbf{J}_{g^\theta}(s)$ is the jacobian of $g^\theta$ evaluated at $s$.

Term ① aligns with our conceptual interpretation of $s$ as the semantic information should remain invariant to any noise information also contained in $z$. Meanwhile the second and third terms correspond to two different sources of error to this interpretation we must potentially take into account. The term ② derives from the encoder struggling to encode samples that were not seen during training, since $\pi(X) \neq p(\hat{X} \mid Z; \theta)p(Z)$. However, provided the reconstructions sufficiently faithful across latent space (i.e. $g^\theta(s) - x$ is small), this term may be ignored. This error can be further mitigated by training the encoder to be more robust with respect to the input (diminishing $\mathbf{J}_{f^\phi}(x)$) with mild additive noise on the input observations or additional regularization terms [54, 52, 58]. Finally, term ③ in equation 6 originates from the decoder having to filter out the stochastic exogenous information from $z$ when decoding. As discussed above in equation 5, the VAE objective already directly minimizes this term by training the decoder output to be invariant to noise samples $u$, thus diminishing $\mathbf{J}_{g^\theta}(s)$.

The bottom line is, as long as the decoder can filter out the exogenous information $u$ from latent samples $z$ and the encoder can recognize the resulting generated samples $\hat{x}$, the endogenous information

$s$ is preserved. We call this process of decoding and reencoding latent samples, $h^{\phi\theta} = f^\phi \circ g^\theta$ the latent response function. Crucially, the latent response function allows us to extract the semantic information from the latent space without knowing how the information is encoded, so although we can identify the structure of the representation, the representation is not necessarily interpretable without ground truth label information. A statistical treatment of this phenomenon is discussed in the appendix, however provided the error terms are sufficiently small the corresponding response distribution $r(\hat{Z} \mid Z; \theta, \phi)$ is shown in equation 7.

$$r(\hat{Z} \mid Z; \theta, \phi) = \mathbb{E}_{p(\hat{X} \mid Z; \theta)} q(\hat{Z} \mid \hat{X}; \phi) \tag{7}$$

One subtlety of our interpretation remains to be addressed: how ambiguities due to overlapping posterior distributions are resolved by latent responses. To match the prior, there will be overlap between posterior distributions which correspond to semantically different samples. Statistically speaking, these ambiguities are captured by the higher moments of $r(\hat{Z} \mid Z; \theta, \phi)$. Despite $\mathbb{E}_{r(\hat{Z} \mid Z; \theta, \phi)}[\hat{S}] \approx S$, the variance can be interpreted to relate to the uncertainty of the encoder in the inference of the endogenous variable from the generated sample, suggesting a potential signal for anomaly detection [55]. From another perspective, a strong deviation between $\hat{S}$ and $S$ implies there is some inconsistency between the encoder and decoder, which identifies the "holes" in the latent space [51], where the encoder has trouble recognizing the samples generated by the decoder.

## 3.1 Interventions

To control the learned generative process, we need to know how to manipulate specific semantic information in the representation. We conceptualize these manipulations as interventions where a chosen latent variable is modified while all others remain unchanged. For example, given latent sample $z$, $\Delta^{(z_j \leftarrow \tilde{z}_j)}(z)$ refers to the interventional sample where the $j$th latent variable is resampled from the marginal of the aggregate posterior $\tilde{z}_j \sim q(Z_j; \phi)$.

When intervening on latent variables $Z$, there are two possible outcomes: either the resulting generated sample is affected significantly, that is to say, the semantic information $S$ is affected by the intervention, or there is no significant change in which case only the noise was affected. In the first case, we identify a specific kind of intervention to manipulate the learned generative process.

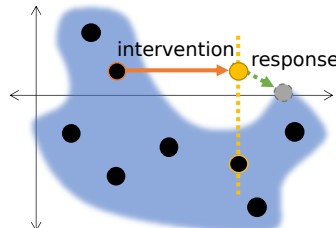

Figure 1: Encoded data points are shown as black points with the surrounding latent manifold shaded blue. An intervention depicted as the orange arrow might replace the semantic information along the horizontal latent dimension with that of the sample with a yellow border. In this case, the intervention results in the latent sample leaving the latent manifold. Now the latent response function approximately filters out the exogenous noise to effective project the sample back onto the manifold. Note that this projection (green arrow) changes both the horizontal and vertical dimensions, from which we infer there is a non-trivial relationship between the horizontal and vertical.

**Latent Response Matrix** This brings us to the first practical tool we introduce based on latent responses, which aims to describe to what extent the latent variables causally affect one another with respect to the learned generative process. Each element in the matrix $\mathbf{M} \in \mathbb{R}^{d*d}$ quantifies the degree to which an intervention in latent variable $j$ causes a response in latent variable $k$ as seen in equation 8, where $h_k^{\theta\phi}$ refers to the $k$th variable of the latent response (see figure 1).

$$M_{jk}^2 = \frac{1}{2} \mathbb{E}_{z \sim p(Z); \tilde{z}_j \sim p(Z_j)} \left[ |h_k^{\phi\theta}(\Delta^{(z_j \leftarrow \tilde{z}_j)}(z)) - h_k^{\phi\theta}(z)|^2 \right] \tag{8}$$

Along the diagonal, $M_{jj}$ can be interpreted as quantifying the extent to which an intervention along the $j$th latent variable is detectable at all. As the value approaches 0, changes in the latent variable do not affect the generated sample in any way detectable by the encoder. For VAEs, this is frequently due to posterior collapse [60, 61, 50, 62]. On the other hand, if the latent variable is maximally informative and the intervention $\tilde{z}_j$ is fully recoverable it implies negligible exogenous noise, so then $h_j^{\phi\theta}(\tilde{z}) \approx \tilde{z}_j$ and $h_j^{\phi\theta}(z) \approx z_j$, and $M_{jj}$ approaches 1, since both $\tilde{z}_j$ and $z_j$ are sampled independently from a standard normal, and the elements are normalized with a factor of $\frac{1}{2}$.

Perhaps even more interesting, the off-diagonal elements of $M_{jk}$ show to what extent an intervention on variable $j$ affects variable $k$. Consequently, the latent response matrix can be interpreted as a weighted adjacency matrix of a directed graph of the learned structural causal graph. Note that there may be cycles in this graph if there is some non-trivial relationship between latent variables. For example, the model could embed a periodic variable into two latent dimensions, such as $j_1$ and $j_2$, to keep the representation continuous. In that case, we would expect an intervention along dimension $j_1$ to elicit a similar response in $j_2$ as the response of $j_1$ from an intervention on $j_2$. Consequently, $M_{j_1 j_2} \approx M_{j_2 j_1} > 0$ suggests $j_1$ and $j_2$ should be treated jointly as a single latent variable. Thus, the latent response matrix not only describes the causal structure of the learned generative process, but it can also identify more complex relationships between latent variables for further analysis.

**Conditioned Response Matrix**  While the latent response matrix $M_{jk}$ lets us quantitatively compare how much each latent dimension affects another, without manual inspection or label information, the correspondence between the learned causal variables the true causal factors is, in general, unknown [19].

However, when we have access to the true generative process, or at least the ground truth labels $y^{(i)} \in \mathcal{Y} \subseteq \mathbb{R}^{d^*}$ corresponding to the observation samples $x^{(i)}$, then there is variant of the latent response matrix, termed the *conditioned response matrix*, to quantify how well the learned variables match with the true ones, which is closely related to the disentanglement of the representation.

The key is to carefully select our interventions such that they only affect one true factor at a time, and then evaluate to what extent these interventions for each of the latent variables are still detectable. Intuitively, if a latent variable $Z_j$ only captures information pertaining to factor $Y_c$, then if we select interventions that only change factor $Y_{c'}$ where $c' \neq c$, then interventions on $Z_j$ do not produce a response, so $M_{jj} \approx 0$.

To condition the set of interventions on a specific factor $Y_c$, we choose a subset of observations which are all semantically identical except for a single factor $Y_c$, $x \sim p(X \mid Y_c, Y_{-c}) p(Y_c) p(Y_{-c})$ where $p(X \mid Y)$ refers to the true generative process given label $Y$ and $Y_{-c}$ refers to all true causal variables except $Y_c$. Then the latent response matrix is computed with interventions exclusively sampled from the resulting aggregate posterior of this subset $q(Z \mid Y_{-c}; \phi) = \int q(Z \mid X; \phi) p(X \mid Y_c, Y_{-c}) p(Y_c) \mathrm{d}X \mathrm{d}Y_c$.

$$M_{cj}^{*\,2} = \frac{1}{2} \mathbb{E}_{z \sim p(Z); \tilde{z}_j \sim q(Z_j \mid Y_{-c}; \phi) p(Y_{-c})} \left[ |h_j^{\phi\theta}(\Delta^{(z_j \leftarrow \tilde{z}_j)}(z)) - h_j^{\phi\theta}(z)|^2 \right] \tag{9}$$

In the context of controllable generation, the conditioned response matrix quantifies how much an intervention on each latent variable can affect each of the true factors of variation. Ideally, each true causal factor would only be manipulable by disjoint subsets of the latent variables, which is commonly referred to as "disentangling" the factors of variation. From the conditioned response matrix, we can identify not only which latent variables contain information pertaining to each of the true factors of variation, but also which factors are affected by an intervention in each of the latent variables.

**Causal Disentanglement Score**  To more easily compare representations, we can aggregate the information in the conditioned response matrix into a single value to measure the degree to which the representation is able to disentangle the true factors of variation. The *causal disentanglement score* (CDS) in equation 10 which allows each latent variable to causally affect a single factor, but penalizes any additional responses. As written, the score is between $\frac{1}{d^*}$ and 1, but we re-scale it to $[0, 1]$.

$$\mathrm{CDS} = \frac{\sum_j \max_c M_{cj}^*}{\sum_{cj} M_{cj}^*} \tag{10}$$

(To our knowledge) none of the existing disentanglement metrics take the learned generative process into account at all. If the task is only to infer the true labels from the observations (as is common), then the decoder is admittedly superfluous, which is why disentanglement methods generally focus on the encoder. However, in tasks such as controllable generation, where disentanglement is obviously valuable, the behavior of the decoder is critical. Here, the main risk in evaluating disentanglement from the encoder alone is that some latent variables may be correlated with true factors of variation without them having any causal effect on those factors when generating new samples. These spurious

correlations also potentially decrease the resulting disentanglement score, which may falsely penalize larger representations.

The conditioned response matrix and associated CDS mirrors the responsibility matrix and disentanglement score introduced by [42]. However, crucially, the responsibility matrix identifies how well each latent variable correlates with each true factor of variation.

**Response Maps**    The last type of analysis we propose focuses on gaining a qualitative understanding of how the data manifold is embedded in the latent space. Specifically, we exploit the ability to probe the latent manifold using the response function to map out the manifolds extent, including estimating its extrinsic curvature. From the definition of $s = f^\phi(x)$ where $s \in \mathcal{Z}$, $x \in \mathcal{X}$ and $f^\phi$ is a smooth deterministic function, we may interpret $s$ as a projection of the data manifold $\mathcal{X}$ into the latent space.

Usually, our analysis of $\mathcal{Z}$ is limited to the observation samples of $x$ we have access, from which manifold learning methods often estimate the intrinsic curvature of the data manifold $\mathcal{X}$. However, using latent responses, we can use any latent sample $z \sim p(Z)$ to probe $\mathcal{Z}$ as long as $\hat{s} \approx s$. Rearranging the terms gives us $u(z)$ in equation 11, whose magnitude can be interpretted as the unsigned distance function to the latent manifold, which is evocative of the neural implicit functions [63, 63, 64] suggesting a variety of further tools we leave for future work.

$$u(z) = s - z \approx h^{\phi\theta}(z) - z \tag{11}$$

Treating $|u(z)|$ as an approximate distance function to the manifold, we compute the mean curvature $H$ of the latent manifold (see appendix for further discussion). In practice, we estimate the necessary gradients by finite differencing across a 2D grid in the latent space, we call the resulting map the *response map*, visualized similarly as in [65].

Qualitatively, with our sign convention, high curvature corresponds to regions where $u(z)$ is small and locally convergent, and consequently where we find the data manifold. Empirically, we also find the divergence of $u(z)$, which is closely related to the mean curvature, and is useful for identifying the regions of the latent space where $u$ diverges, which may be interpreted as possible "holes" in the latent space.

When interpolating between latent samples $z$, ideally we would follow the geodesic of the data manifold. This can be done by estimating the Riemannian metric from the data samples and integrating an expensive ODE to find the geodesic connecting samples [25]. The Riemmanian metric relates to the intrinsic curvature of the data manifold, which is independent of the embedding, consequently guaranteeing we find the geodesic independent of the representation. However, in our case, we use the latent responses to estimate the extrinsic curvature which does depend on the embedding, so the resulting path may not be optimal with respect to the underlying data manifold. Nevertheless, we optimize the path along the response map to stay in high curvature regions, thereby effectively are finding a path in the latent space which stays near the data manifold.

## 4   Toy Example: The Double Helix

To illustrate how the latent response framework can be used to study the representation learned by a VAE, we show the process when learning a 2D representation for samples from a double helix embedded in $\mathbb{R}^3$. Disregarding the additive noise, the data manifold has two degrees of freedom (data manifold formally defined in the appendix).This analysis is largely independent of the precise neural network architecture, provided the model has sufficient capacity to learn a satisfactory representation (hyperparameters in the appendix).

Figure 2 provides an example for how the response maps can be used to trace the latent representation through the mean curvature and divergence. Note that the magnitude of the response $|u(z)|$ is not sufficient to identify the latent manifold since both the regions with maximal and minimal curvature have a minimal response. The divergence shows most of the latent space has a slightly negative divergence, implying most of the latent space converges, rather than diverges, which is consistent with expectations. The mean curvature shows what regions of the latent space the map converges to in yellow, from which we can recognize the structure of the learned latent manifold. Finally, the right most plot shows the aggregate posterior $q(S \mid X; \phi)$ of the 1024 training samples.

This toy example also motivates the value of meaningful interpolations as seen in figure 3. The path in red shows the shortest euclidean path between the two samples in orange, but note that the



Figure 2: Depicted are three quantities derived using the latent response function compared to the aggregate posterior shown on the far right for the representation learned by a VAE for the double helix toy dataset. Note that almost all the density of the posterior is in regions that have positive curvature, corresponding to the data manifold.

corresponding path in the observation space jumps from one strand to another twice. Meanwhile, the path that maximizes the estimated mean curvature is shown in green and produces a much more reasonable interpolation.

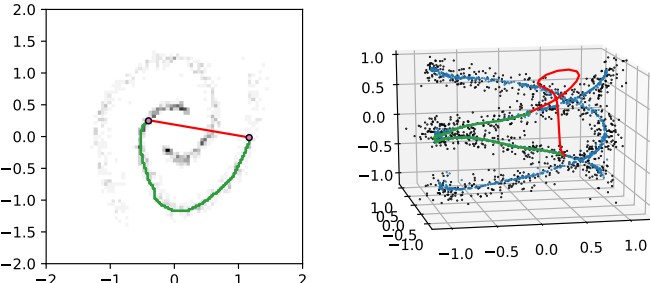

Figure 3: The left plot shows the 2D latent space including the aggregate posterior density in black, and two possible interpolations between the two pink points. Meanwhile, the plot on the right shows the ambient space with the black points being the observed data samples, with the blue points showing the reconstructed samples, and the paths in the ambient space corresponding to the ones in the latent space. Note how the path in green follows the learned manifold and consequently much more consistent in the ambient space compared to the shortest euclidean path in red.

## 5   Experiments & Results

**Experimental Setup**   We apply our new tools on a small selection of common benchmark datasets, including 3D-Shapes [66], MNIST [67], and Fashion-MNIST [68] [3]. Our methods are directly applicable to any VAE-based model, and can readily be extended to any autoencoders. Nevertheless, we focus our empirical evaluation on vanilla VAEs and some $\beta$-VAEs (denoted by substituting the $\beta$, so 4-VAE refers to a $\beta$-VAE where $\beta = 4$). Specifically, here we mostly analyze a 4-VAE model with a $d = 24$ latent space trained on 3D-Shapes (except for table 1) referred to as Model A, and include results on a range of other models in the appendix. All our models use four convolution and two fully-connected layers in the encoder and decoder. The models are trained using Adam [69] with a learning rate of 0.001 for 100k steps (see appendix for details).

**Qualitative Understanding**   The most direct way to get a better understanding of the manifold structure is the visualization of the response maps, in particular with the mean curvature (see figure 4). Unfortunately, since the curvature is estimated numerically using a grid of samples, the maps do not scale well to the whole latent space. Here the latent response matrices help identify pairs of related latent dimensions, which can then be analyzed more closely with a response map. Furthermore, currently, all these response maps are aligned to the axes of the latent space. Although VAEs do align information along the axes somewhat [70], any off-axis structure is missed since the off-axes responses are completely ignored. Since, an implicit requirement of disentangled representations is that the information is axis-aligned [19], the response maps present the most striking results for disentangled representations (see appendix for more examples).

---

[3]All are provided with an MIT, Apache or Creative Commons License

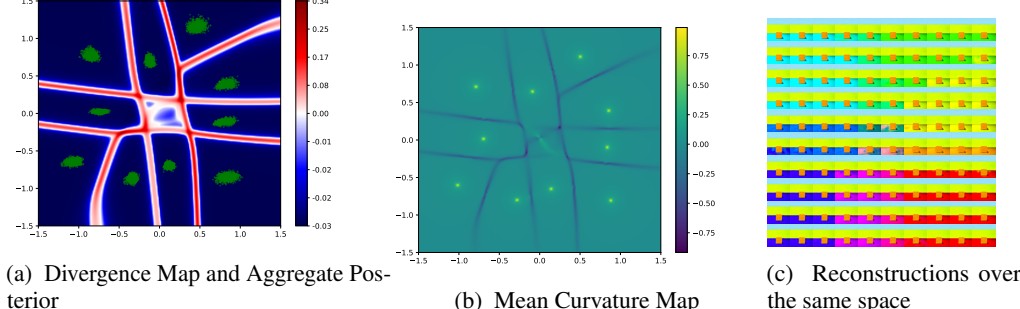

(a) Divergence Map and Aggregate Posterior

(b) Mean Curvature Map

(c) Reconstructions over the same space

Figure 4: A projection along dimensions 16 (horizontally) and 22 for Model A (4-VAE) shows the computed divergence of the response field in blue and red while the green points are samples from the aggregate posterior. 4b shows the mean curvature, which identifies 10 points where the curvature spikes and the boundaries between the regions corresponding to different clusters in the posterior. Finally, from the corresponding reconstructions in 4c (with all other latent variables fixed) it becomes clear that each of the clusters in the posterior corresponds to a different floor hue. Note that, although the aggregate posterior is highly concentrated at a few points, the negative divergence almost everywhere suggests the extent of $U \mid X$ the decoder can handle extends well beyond the posterior (as confirmed by the reconstructions).

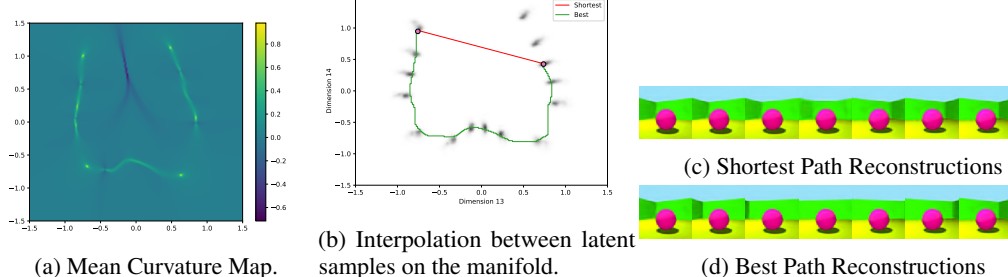

(a) Mean Curvature Map.

(b) Interpolation between latent samples on the manifold.

(c) Shortest Path Reconstructions

(d) Best Path Reconstructions

Figure 5: Using Model A (4-VAE) in a similar setting to figure 3, we now search for an interpolation between two latent samples from the posterior along the latent manifold, visualized in 5a. Figure 5b compares this best path (in green) to the shortest path in euclidean distance (in red), Finally, latent samples along each of the paths at even intervals are decoded showing that the shortest path results in blurry, unrealistic shadows in the middle of 5c compared to 5d.

From the mean curvature response map in figure 5a, we see that the manifold is particularly nonlinear along these two latent dimensions (13 and 14). Consequently, there can be a dramatic difference between the geodesic (here approximated using the mean curvature), compared to the shortest path in euclidean space as seen in figure 5.

**Causal Disentanglement** Table 1, unsurprisingly, shows the disentanglement increasing with increasing $\beta$. Our proposed CDS score correlates strongly with the other disentanglement metrics. Perhaps noteworthy is that even though the CDS and DCI-D scores are computed in similar ways (the vital difference being whether responsibility rests on a causal link or a statistical correlation), the DCI-D scores are consistently lower than the CDS scores. This may be explained by the DCI-D metric

| Name | CDS | DCI-D | IRS | MIG |
|------|-----|-------|-----|-----|
| 1-VAE | 0.44 | 0.3 | 0.44 | 0.07 |
| 2-VAE | 0.49 | 0.36 | 0.46 | 0.09 |
| 4-VAE | 0.58 | 0.46 | 0.51 | 0.16 |
| 8-VAE | 0.71 | 0.66 | 0.63 | 0.21 |
| 1-VAE | 0.52 | 0.49 | 0.48 | 0.09 |
| 2-VAE | 0.61 | 0.58 | 0.52 | 0.15 |
| 4-VAE | 0.72 | 0.67 | 0.57 | 0.17 |
| 8-VAE | 0.78 | 0.73 | 0.64 | 0.2 |

Table 1: Comparing disentanglement metrics for $\beta$-VAEs trained on 3D-shapes with varying $\beta$. For the models in the first four rows $d = 12$, while $d = 24$ for the remaining four. While the CDS generally correlates well with other metrics, notably, the DCI-D score is consistently slightly lower which may be due to spurious correlations between latent variables and the true factors.

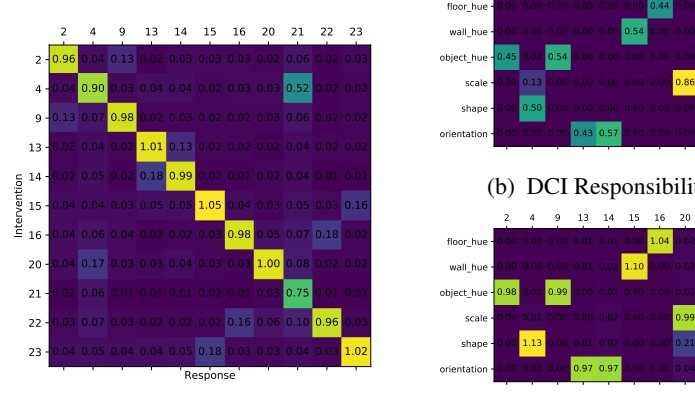

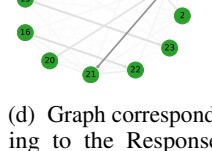

(b) DCI Responsibility Matrix

(a) Latent Response Matrix

(c) Conditioned Response Matrix

(d) Graph corresponding to the Response Matrix

Figure 6: These are the (6a) latent response matrix, (6b) DCI Responsibility matrix [42], (6c) conditioned response matrix, and the (6d) derived from the latent response matrix for Model A (4-VAE) ($d = 24$) trained on 3D-Shapes. The responsibility matrix shows the predictability of each latent dimension (column) for each factor of variation (row), while the conditioned response matrix measures the effect an intervention in each latent dimension (column) has provided that the intervention can changes a specific factor of variation (row). Note that from the conditional response matrix we see only dimensions 16 and 22 are causally linked to the floor hue, for which the structure is further visualized in figure 4.

taking additional spurious correlations between latent variables into account (as seen in figure 6), while the CDS metric focuses on the causal links, so the DCI-D has an undeservedly low score.

**Causal Structure**  Closer comparison between the conditioned response matrices and the responsibility matrices reveal more how the DCI-D metric and CDS differ. Figure 6 shows the responsibility matrix matches the conditioned response matrix for the most part. The only exception being latent variables 4 and 20. Since the DCI framework identifies which latent variables are most predictive for the true factors, it cannot distinguish between a correlation and a causal link. In this case, the DCI metric recognized a correlation between dimension 4 and the "scale" factor. However, from the latent response matrix (and the graph 6d) we see interventions on dim 20 have a significant effect on dim 4, but not vice versa. Consequently, we identify dim 20 is a parent of dim 4 in the learned causal graph. Since dim 20 is closely related with the "scale" factor the causal link to dim 4 results in dim 4 being correlated with "scale". The conditioned response matrix correctly identifies that it is dimension 20 which primarily affects the scale, but indirectly also affects the shape through dimension 4. This is a prime example of how causal reasoning can avoid misattributing responsibility due to spurious correlations.

## 6  Conclusion

In this work, we have introduced and motivated the latent response framework including a variety of tools to better visualize and understand the representations learned by variational autoencoders. Given an intervention on a sample in the latent space, the latent response quantifies the degree to which that intervention affects the semantic information in the sample. Therefore, we can think of this analysis as leveraging the *interventional consistency* of a representation to study the geometric and causal structure therein. Notably, the current analysis relies on a certain degree of axis-aligned structure in the latent space, which makes these tools especially useful for understanding the structure of disentangled representation. Another limitation is that computing the latent response maps to, for example, improve interpolations, does not scale well for the large representations of high fidelity generative models [8]. Consequently, our experiments thus far have focused on synthetic datasets designed for evaluating disentanglement methods. However, latent responses do not require any ground truth label information, which is particularly promising for better understanding representations of real datasets and consequently speeding up development of not just better performing representation learning techniques, but also more interpretable and trustworthy [71] models.

## Acknowledgements

This work was supported by the German Federal Ministry of Education and Research (BMBF): Tübingen AI Center, FKZ: 01IS18039B, and by the Machine Learning Cluster of Excellence, EXC number 2064/1 – Project number 390727645. The authors thank the International Max Planck Research School for Intelligent Systems (IMPRS-IS) for supporting Felix Leeb.

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
