# A  Appendix

## A.1  Latent Responses

In a similar setup as [54], we can extend the joint distribution $p(X, Z)$ to include the reconstruction and response as $p(X, Z, \hat{X}, \hat{Z})$, where crucial question is how the posterior $q(Z \mid X; \phi)$ relates to the response $q(\hat{Z} \mid \hat{X}; \phi)$ where $Z$ and $\hat{Z}$ are related by (also shown in equation 7):

$$r(\hat{Z} \mid Z; \theta, \phi) = \int q(\hat{Z} \mid \hat{X}; \phi) p(\hat{X} \mid Z; \theta) \mathrm{d}\hat{X} \tag{12}$$

Note, that $r(\hat{Z} \mid Z; \theta, \phi)$ is equivalent to the transition kernel $\mathcal{Q}_{\text{AVAE}}$ in [54]. However, crucially, we do not make two assumptions used to derive the AVAE objective. Firstly, we do not assume that the decoder is a one-to-one mapping between latent samples and a corresponding generated sample. The contractive behavior observed in the latent space of autoencoders [59], suggests a many-to-one mapping is more realistic, which may be interpretted as the decoder filtering out useless exogenous information from the latent code. Consequently, we also do not treat $p(\hat{Z}; \theta, \phi) = \mathbb{E}_{p(Z)}[r(\hat{Z} \mid Z; \theta, \phi)]$ as a normal distribution, which would imply the encoder perfectly inverts the decoder.

Consider the reconstructions $\hat{X}$ of the maximally overfit encoder $q(z = Z \mid x_i = X; \tilde{\phi}) = \delta(s_i - z)$ (recall $s_i = f^{\phi}(x_i)$) and decoder $p(\hat{X} \mid Z; \tilde{\theta})$. Since the autoencoder is trained on the empirical generative process $\pi(X)$ rather than the true generative process $p(X)$, the overfit decoder generates samples from $p(\hat{X}; \tilde{\theta}) = \int p(\hat{X} \mid Z; \tilde{\theta}) p(Z) \mathrm{d}Z = \pi(X)$, which does not have continuous support. For such a decoder, all exogenous noise is completely removed and the decoder mapping is obviously many-to-one, and it follows that $r(\hat{Z} = \hat{z} \mid Z = z; \tilde{\theta}, \tilde{\phi}) = \delta(\hat{z} - s)$ (recall $z = s + u$).

Now consider the more desirable (and perhaps slightly more realistic) setting where the autoencoder extrapolates somewhat beyond $\pi(X)$ to resemble $p(X)$, in which case decoding the latent sample $z \sim q(Z \mid X = x; \phi)$ to generate $\hat{x} \sim p(\hat{x} = \hat{X} \mid z = Z; \theta)$ will not necessarily match the observation $x$, which, by our definition of endogenous information, implies a change in the endogenous information contained in $z$. When re-encoding to get $q(\hat{Z} \mid \hat{X} = \hat{x}; \phi)$, the changes in the endogenous information result in some width to the distribution over $\hat{Z}$.

## A.2  Derivation of Equation 6

Starting from our definition of $\hat{s} = f^{\phi}(\hat{x})$ where $\hat{x} = g^{\theta}(z)$, $z = f^{\phi}(x)$, $z = s + u$, and $\epsilon = \hat{x} - x$. The high-level goal is expand $f^{\phi}$ around $x$ and then $g^{\theta}$ around $s$ to first order.

$$\hat{s} = f^{\phi}(\hat{x})$$

$$\hat{s} = f^{\phi}(x + \epsilon)$$

$$\hat{s} = f^{\phi}(x) + \mathbf{J}_{f^{\phi}}(x)\epsilon + \mathcal{O}\left[\epsilon^2\right]$$

$$\hat{s} = f^{\phi}(x) + \mathbf{J}_{f^{\phi}}(x)\left[g^{\theta}(z) - x\right] + \mathcal{O}\left[\epsilon^2\right]$$

$$\hat{s} = f^{\phi}(x) + \mathbf{J}_{f^{\phi}}(x)\left[g^{\theta}(s + u) - x\right] + \mathcal{O}\left[\epsilon^2\right]$$

$$\hat{s} = f^{\phi}(x) + \mathbf{J}_{f^{\phi}}(x)\left[g^{\theta}(s) + \mathbf{J}_{g^{\theta}}(s)u - x\right] + \mathcal{O}\left[\epsilon^2\right] + \mathcal{O}\left[u^2\right]$$

$$\hat{s} = f^{\phi}(x) + \mathbf{J}_{f^{\phi}}(x)(g^{\theta}(s) - x) + \mathbf{J}_{f^{\phi}}(x)\mathbf{J}_{g^{\theta}}(s)u + \mathcal{O}\left[\epsilon^2\right] + \mathcal{O}\left[u^2\right]$$

### A.3 Comparing the Conditioned Response Matrix and the DCI Responsibility Matrix

In [42], the responsibility matrix is used to evaluate the disentanglement of a learned representation. In the matrix, element $R_{ij}$ corresponds to the relative importance of latent variable $j$ in predicting the true factor of variation $i$ for a simple classifier trained with full supervision to recover the true factors from the latent vector. Although the scalar scores (DCI-d and CDS) are computed identically from the respective matrices, there are important practical and theoretical distinctions in the DCI and latent response frameworks.

First and most importantly, since the DCI framework only uses the encoder, the learned generative process is not taken into account at all. Consequently, the DCI framework (and other existing disentanglement metrics) fail to evaluate how disentangled the causal drivers of the learned generative process are, and instead evaluate which latent variables are correlated with true factors. Furthermore, practically speaking, the DCI framework is sensitive to a variety of hyperparameters such as the exact design and training of the model [72], while the conditioned response matrix has far fewer (and more intuitive) hyperparameters relating to the Monte carlo integration.

Interestingly, the DCI responsibility matrices do often resemble the conditioned response matrices, suggesting that relying on correlations instead of a full causal analysis, can yield similar results. Obviously as the data becomes more challenging and realistic, and the true generative process involves a more complicated causal structure, then we may expect the DCI responsibility matrix to become less reliable for analyzing the generative model structure. In fact, then the learned causal structure estimated using the latent response matrix may be used in tandem to develop a structure-aware disentanglement metric.

### A.4 Mean Curvature for Manifold Learning

The geometry of learned representations with a focus on the generalization ability of neural networks has been discussed in [73]. One key problem is that the standard Gaussian prior used in variational autoencoders relies on the usual Lebesgue measure which in turn, assumes a Euclidean structure over the latent space. This has been demonstrated to lead to difficulties in particular when interpolating in the latent space [25, 74, 75] due to a manifold mismatch [76, 77]. Given the complexity of the underlying data manifold, a viable alternative is based on riemanian geometry [78] which has previously been investigated for alternative probabilistic models like Gaussian Process regression [79].

These methods focus on the intrinsic curvature of the data manifold, which does not depend on the specific embedding of the manifold in the latent space. However, our focus is precisely on how the data manifold is embedded in the latent space, to (among other things) quantify the relationships between latent variables and how well the representation disentangles the true factors of variation. Consequently, we focus on the extrinsic curvature, and more specifically the mean curvature which can readily be estimated using the response maps.

As discussed in the main paper, $|u(z)| = |z - s|$ is interpreted as a distance where $|u(z)| = 0$ implies $z$ is on the latent manifold and there is no exogenous noise. The gradient of this function $\nabla_z |u(z)|$, effectively projects any point in the latent space onto the endogenous manifold. Similarly, the mean curvature (equation 13) can be computed, which can be interpreted as identifying the regions in the latent space where the $|u(z)|$ converges and diverges. These gradients are estimated numerically by finite differencing.

$$H = -\frac{1}{2}\nabla_z \cdot \left( \frac{\nabla_z |u(z)|}{|\nabla_z |u(z)||} \right) = -\frac{1}{2}\nabla_z \cdot \frac{u(z)}{|u(z)|} \tag{13}$$

### A.5 Double Helix Example Details

To illustrate how the latent response framework can be used to study the representation learned by a VAE, we show the process when learning a 2D representation for samples from a double helix embedded in $\mathbb{R}^3$, defined as:

$$x_i = [A_1 \cos(\pi(\omega t_i + n_i)), A_2 \sin(\pi(\omega t_i + n_i)), A_3 t_i]^T + \epsilon_i \tag{14}$$

where $t_i \sim \text{Uniform}(-1, 1)$, $n_i \sim \text{Bernoulli}(0.5)$, $\epsilon_i \sim \mathcal{N}(\mathbf{0}, \sigma\mathbf{I})$. For this experiment, we set $A_1 = A_2 = A_3 = \omega = 1$ and $\sigma = 0.1$.

Disregarding the additive noise $\epsilon_i$, the data manifold has two degrees of freedom, which are the strand location $t_i$ and the strand number $n_i$.

To provide the model sufficient capacity, we use four hidden layers with 32 units each for the encoder and decoder. We train until convergence (at most 5k steps) with $\beta = 0.05$ using an Adam optimizer on a total of $N = 1024$ training samples (see the supplementary code for the full training and evaluation details).

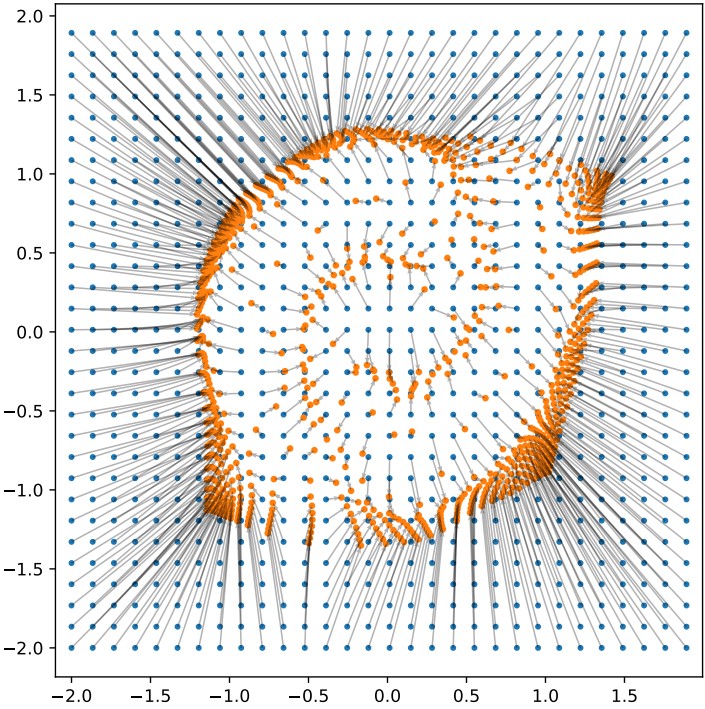

Figure 7: The response map of the representation trained on the double helix. Starting from the latent samples (blue dots), applying the decoder followed by the encoder (i.e. response function) results in the orange dots connected by the black arrows. Note that applying the response function effectively contracts points all over the latent space into a relatively small non-linear region, corresponding to endogenous information.

## A.6 Architecture and Training Details

All our models are based on the same convolutional neural network architecture detailed in table 10 so that in total models have approximately 500k trainable parameters. For the smaller datasets MNIST and Fashion-MNIST, samples are upsampled to 32x32 pixels from their original 28x28 and the one convolutional block is removed from both the encoder and decoder.

The datasets are split into a 70-10-20 train-val-test split, and are optimized using Adam [69] with a learning rate of 0.0001, weight decay 0, and $\beta_1$, $\beta_2$ of 0.9 and 0.999 respectively. The models are trained for 100k iterations with a batch size of 64 (128 for MNIST and Fashion-MNIST).

| Input 64x64x3 image |
| --- |
| Conv Layer (64 filters, k=5x5, s=1x1, p=2x2) |
| Max pooling (filter 2x2, s=2x2) |
| Group Normalization (8 groups, affine) |
| ELU activation |
| Conv Layer (64 filters, k=3x3, s=1x1, p=1x1) |
| Max pooling (filter 2x2, s=2x2) |
| Group Normalization (8 groups, affine) |
| ELU activation |
| Conv Layer (64 filters, k=3x3, s=1x1, p=1x1) |
| Max pooling (filter 2x2, s=2x2) |
| Group Normalization (8 groups, affine) |
| ELU activation |
| Conv Layer (64 filters, k=3x3, s=1x1, p=1x1) |
| Max pooling (filter 2x2, s=2x2) |
| Group Normalization (8 groups, affine) |
| ELU activation |
| Conv Layer (64 filters, k=3x3, s=1x1, p=1x1) |
| Max pooling (filter 2x2, s=2x2) |
| Group Normalization (8 groups, affine) |
| ELU activation |
| Fully-connected Layer (256 units) |
| ELU activation |
| Fully-connected Layer (128 units) |
| ELU activation |
| Fully-connected Layer ($2d$ units) |
| Output posterior $\mu$ and $\log \sigma$ |

Figure 8: Encoder Architecture

| Input $d$ latent vector |
| --- |
| Fully-connected Layer (128 units) |
| ELU activation |
| Fully-connected Layer (256 units) |
| ELU activation |
| Fully-connected Layer (256 units) |
| ELU activation |
| Bilinear upsampling (scale 2x2) |
| Conv Layer (64 filters, k=3x3, s=1x1, p=1x1) |
| Group Normalization (8 groups, affine) |
| ELU activation |
| Bilinear upsampling (scale 2x2) |
| Conv Layer (64 filters, k=3x3, s=1x1, p=1x1) |
| Group Normalization (8 groups, affine) |
| ELU activation |
| Bilinear upsampling (scale 2x2) |
| Conv Layer (64 filters, k=3x3, s=1x1, p=1x1) |
| Group Normalization (8 groups, affine) |
| ELU activation |
| Bilinear upsampling (scale 2x2) |
| Conv Layer (64 filters, k=3x3, s=1x1, p=1x1) |
| Group Normalization (8 groups, affine) |
| ELU activation |
| Conv Layer (3 filters, k=3x3, s=1x1, p=1x1) |
| Sigmoid activation |
| Output 64x64x3 image |

Figure 9: Decoder Architecture

Figure 10: Model architectures where "k" is the kernel size, "s" is the stride, and "p" is the zero-padding

# B Additional Results

## B.1 3D-Shapes

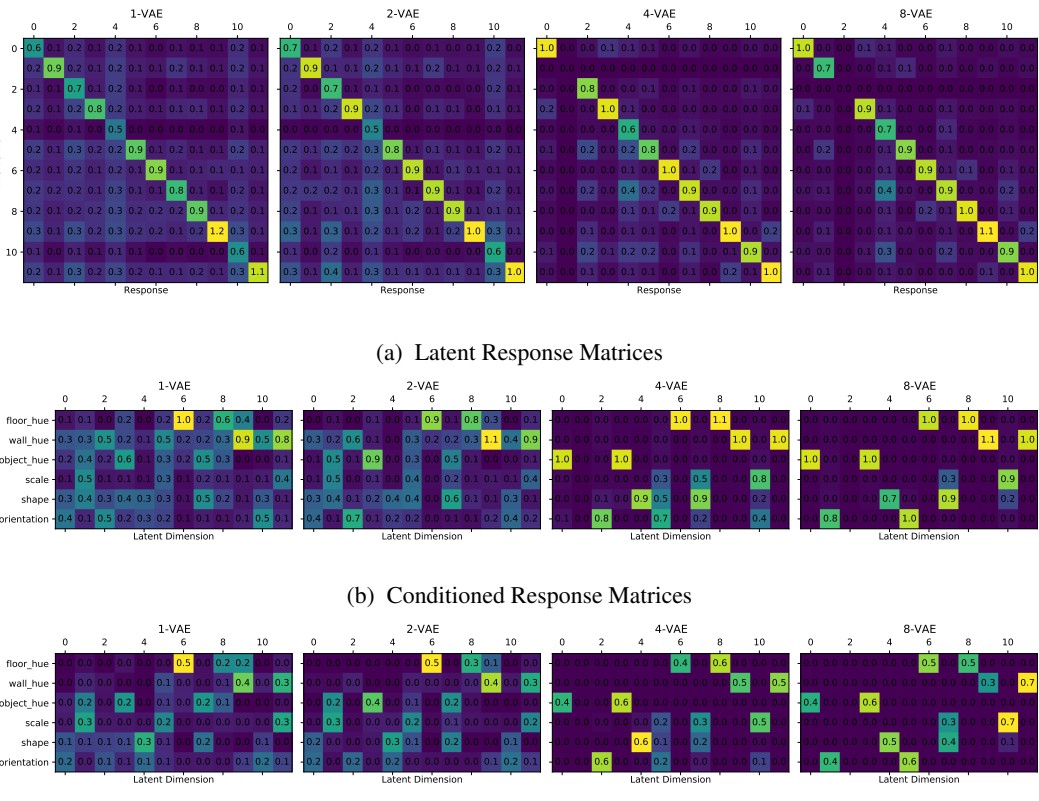

(a) Latent Response Matrices

(b) Conditioned Response Matrices

(c) DCI Responsibility Matrices

Figure 11: Response and Responsibility matrices for several VAEs ($d = 12$).

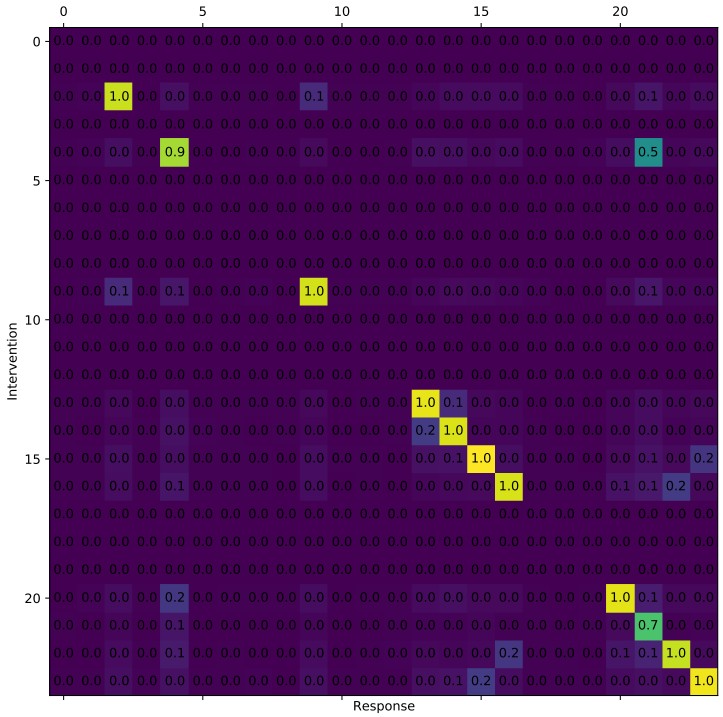

(a) Latent Response Matrix

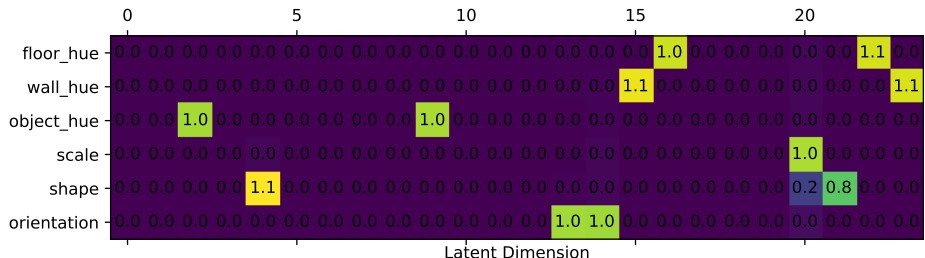

(b) Conditioned Response Matrix

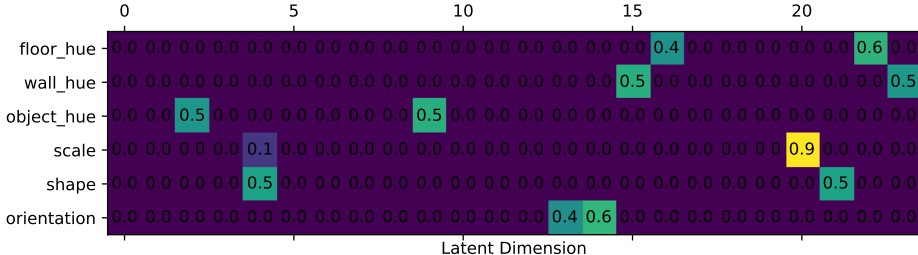

(c) DCI Responsibility Matrix

Figure 12: Full response and responsibility matrices of the 4-VAE ($d = 24$) also shown in figure 6. Note how the Latent Response matrices (12a) shows a categorical difference between the latent dimensions where the diagonal element is close to zero (non-causal), compared to the dimensions with diagonal elements close to 1 (causal).

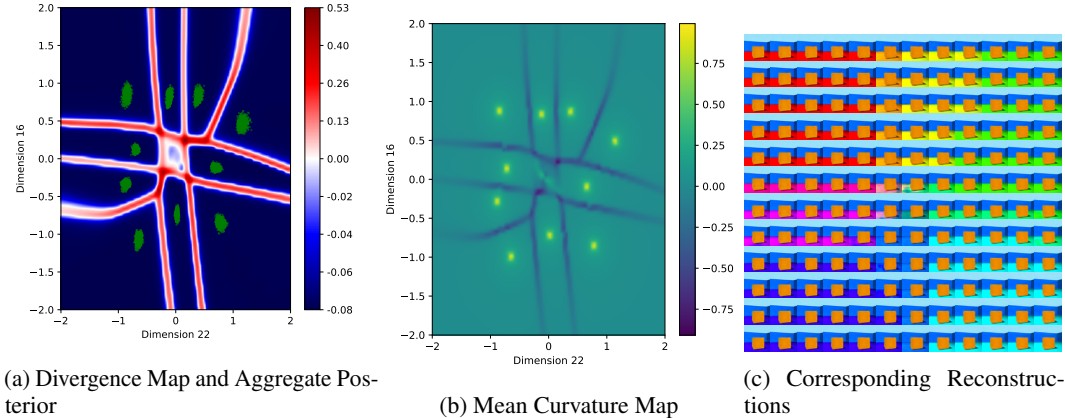

(a) Divergence Map and Aggregate Posterior

(b) Mean Curvature Map

(c) Corresponding Reconstructions

Figure 13: Visualization of the representation learned by a 4-VAE trained on 3D-Shapes (same model as in figure 12).

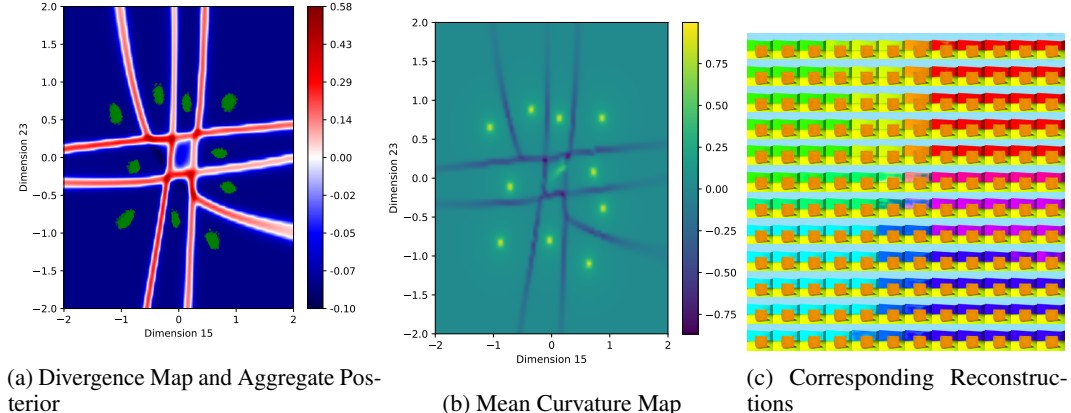

(a) Divergence Map and Aggregate Posterior

(b) Mean Curvature Map

(c) Corresponding Reconstructions

Figure 14: Visualization of the representation learned by a 4-VAE trained on 3D-Shapes (same model as in figure 12).

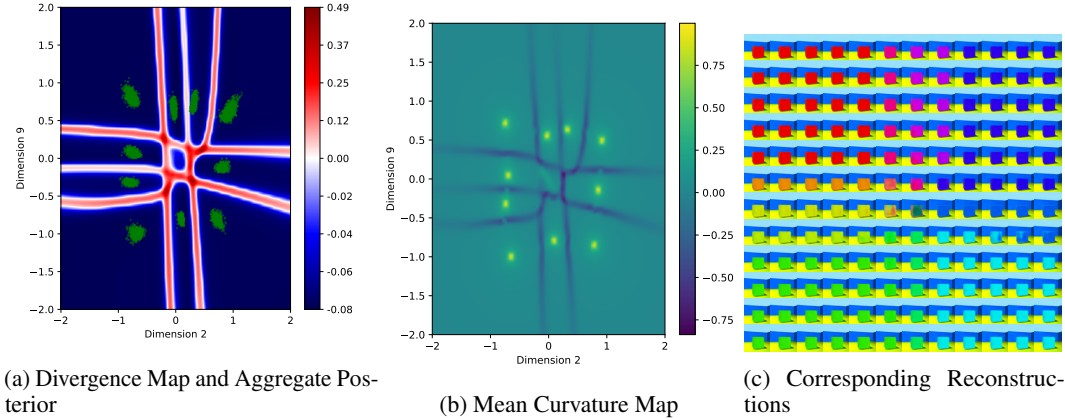

(a) Divergence Map and Aggregate Posterior

(b) Mean Curvature Map

(c) Corresponding Reconstructions

Figure 15: Visualization of the representation learned by a 4-VAE trained on 3D-Shapes (same model as in figure 12).

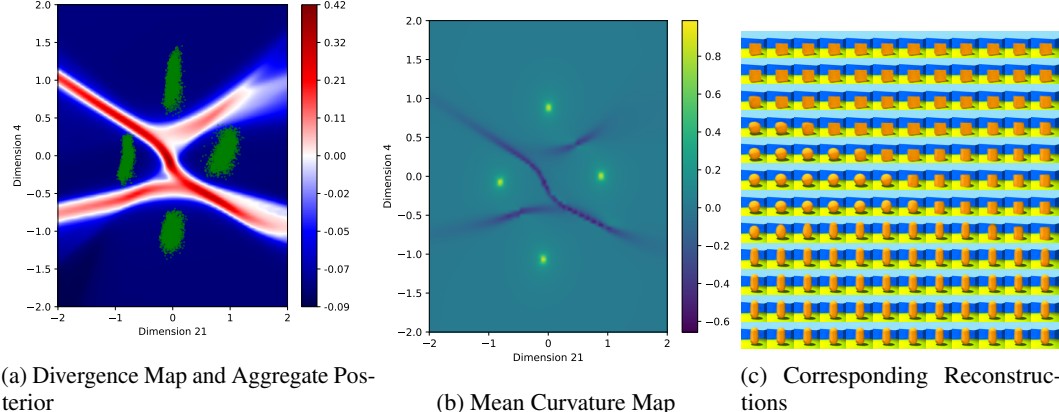

(a) Divergence Map and Aggregate Posterior

(b) Mean Curvature Map

(c) Corresponding Reconstructions

Figure 16: Visualization of the representation learned by a 4-VAE trained on 3D-Shapes (same model as in figure 12). This projection is particularly interesting as the information encoding shape is not exactly axis-aligned, leading to a slight mismatch between the aggregate posterior and the divergence maps. As our visualizations are presently confined to two dimensions, the structure can become significantly more obscured to us if the information is not disentangled and axis-aligned.

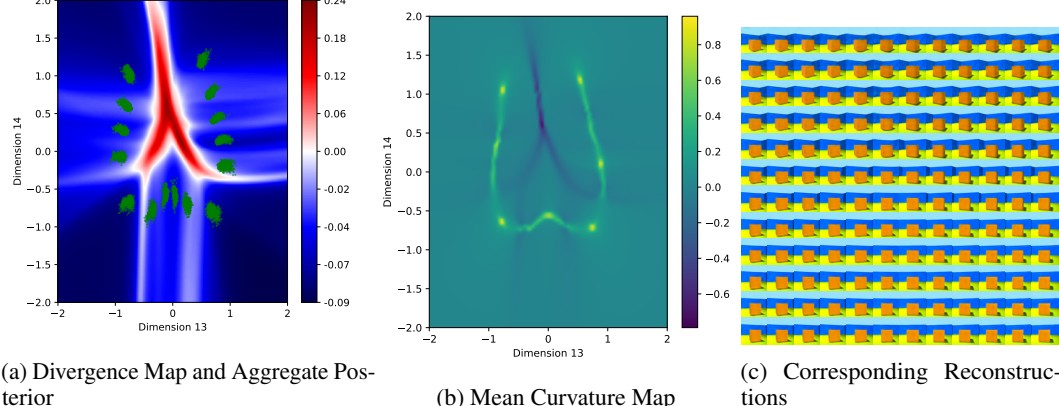

(a) Divergence Map and Aggregate Posterior

(b) Mean Curvature Map

(c) Corresponding Reconstructions

Figure 17: Visualization of the representation learned by a 4-VAE trained on 3D-Shapes (same model as in figure 12).

## B.2 MPI3D

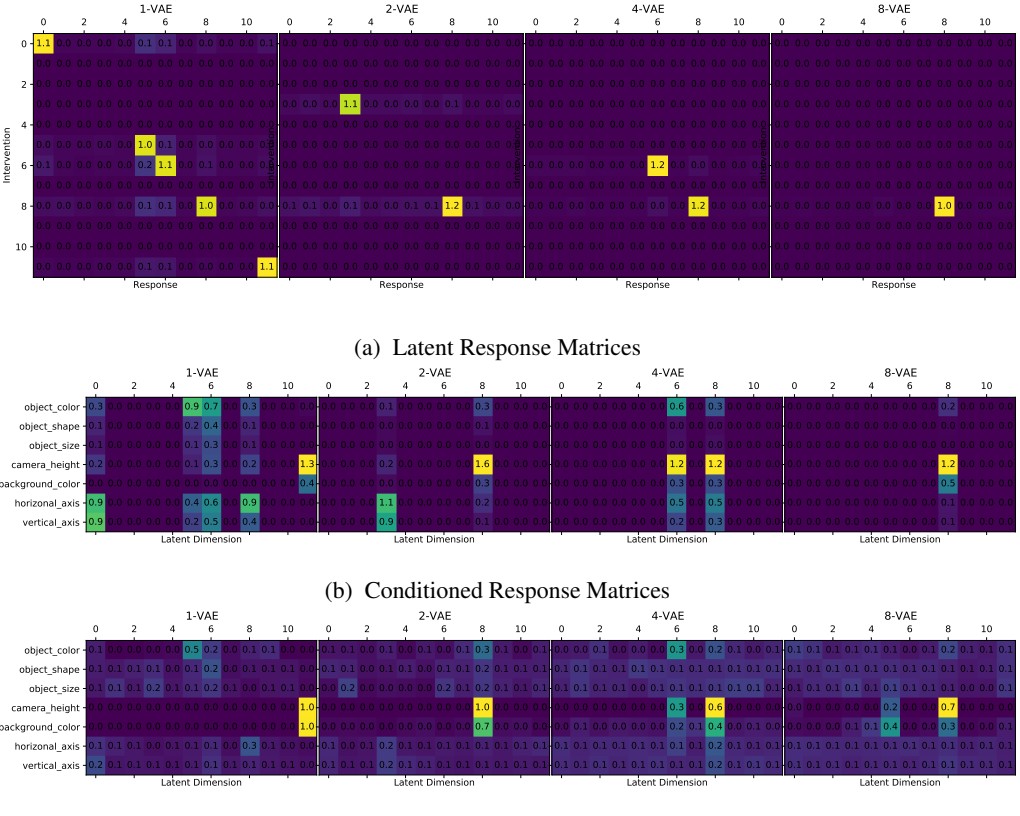

(a) Latent Response Matrices

(b) Conditioned Response Matrices

(c) DCI Responsibility Matrices

Figure 18: Response and Responsibility matrices for several VAEs ($d = 12$) trained on the MPI3D Toy dataset.

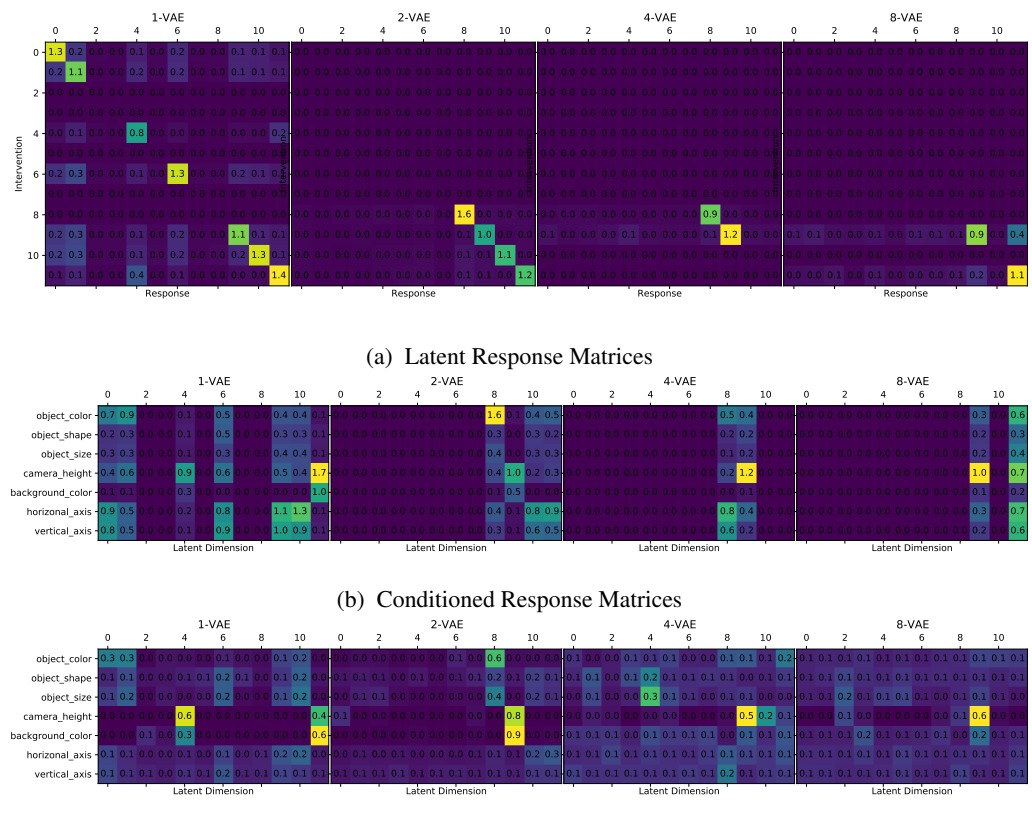

(a) Latent Response Matrices

(b) Conditioned Response Matrices

(c) DCI Responsibility Matrices

Figure 19: Response and Responsibility matrices for several VAEs ($d = 12$) trained on the MPI3D Real dataset.

| Name | CDS | DCI-D | IRS | MIG |
|---|---|---|---|---|
| 1-VAE | 0.69 | 0.33 | 0.58 | 0.32 |
| 2-VAE | 0.86 | 0.17 | 0.59 | 0.14 |
| 4-VAE | 0.66 | 0.11 | 0.61 | 0.05 |
| 8-VAE | 1 | 0.13 | 0.79 | 0.1 |
| 1-VAE | 0.61 | 0.24 | 0.51 | 0.07 |
| 2-VAE | 0.69 | 0.26 | 0.72 | 0.24 |
| 4-VAE | 0.4 | 0.09 | 0.75 | 0.04 |
| 8-VAE | 0.7 | 0.08 | 0.71 | 0.04 |

Table 2: disentanglement scores for the MPI3D Toy (first four rows) and Real (last four rows).

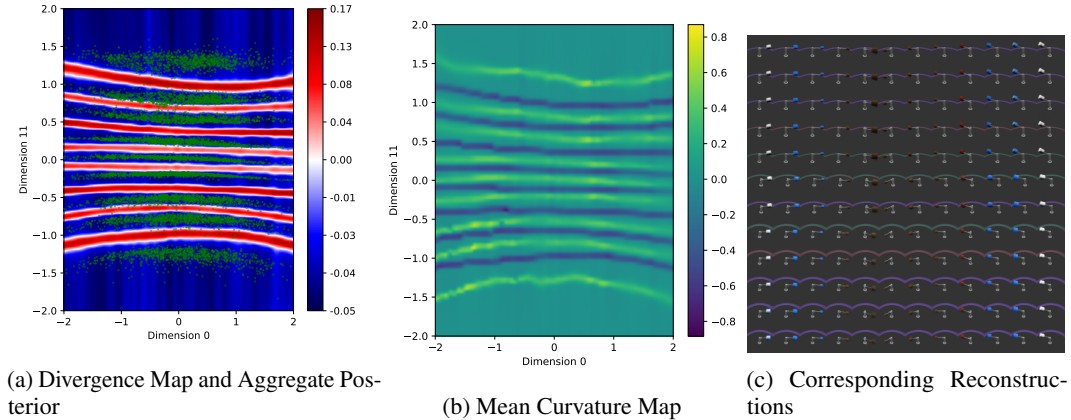

(a) Divergence Map and Aggregate Posterior

(b) Mean Curvature Map

(c) Corresponding Reconstructions

Figure 20: Visualization of the representation learned by the 1-VAE trained on MPI3D Toy.

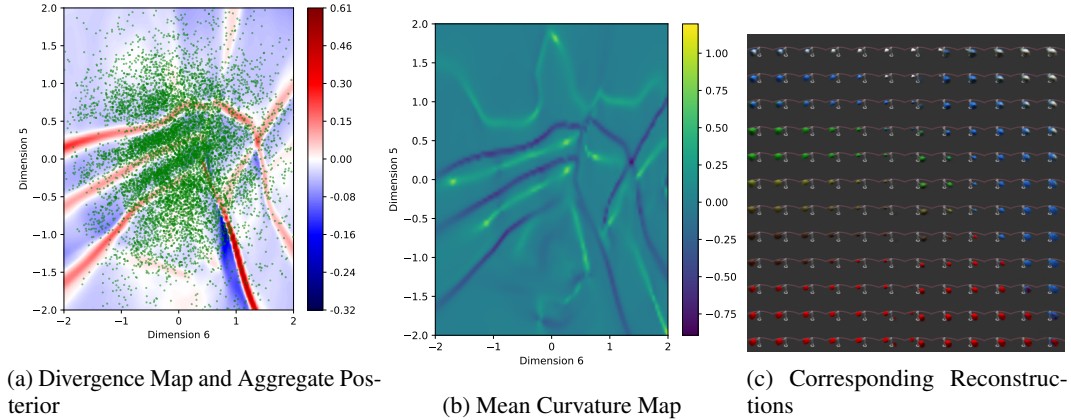

(a) Divergence Map and Aggregate Posterior

(b) Mean Curvature Map

(c) Corresponding Reconstructions

Figure 21: Visualization of the representation learned by the 1-VAE trained on MPI3D Toy.

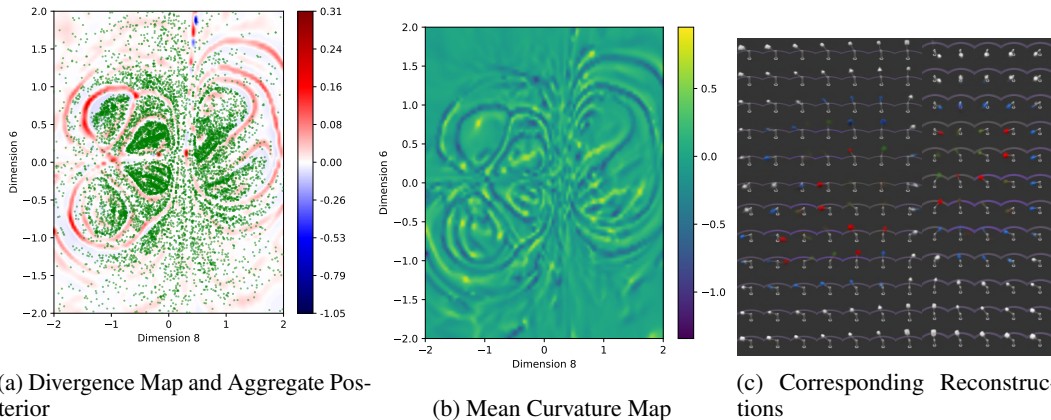

(a) Divergence Map and Aggregate Posterior

(b) Mean Curvature Map

(c) Corresponding Reconstructions

Figure 22: Visualization of the representation learned by the 4-VAE trained on MPI3D Toy. Note that due to posterior collapse, the full latent manifold is contained in this projection (see the corresponding response matrix in figure 18).

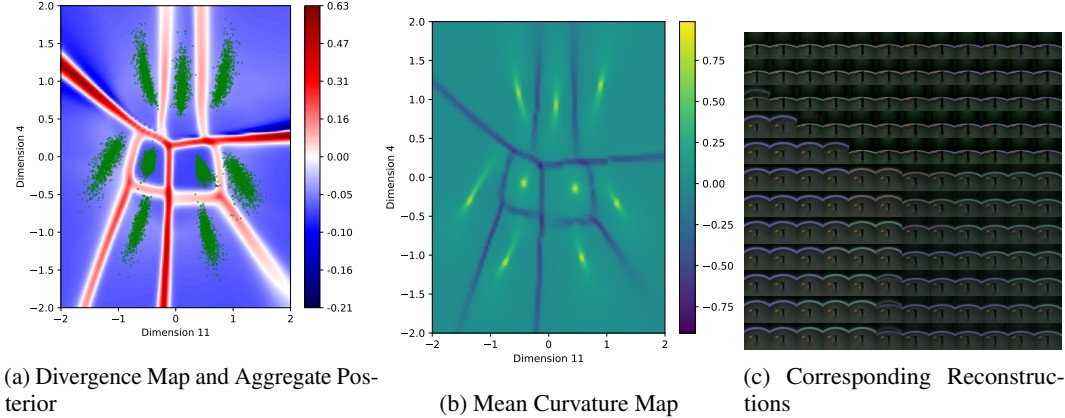

(a) Divergence Map and Aggregate Posterior

(b) Mean Curvature Map

(c) Corresponding Reconstructions

Figure 23: Visualization of the representation learned by the 1-VAE trained on MPI3D Real.

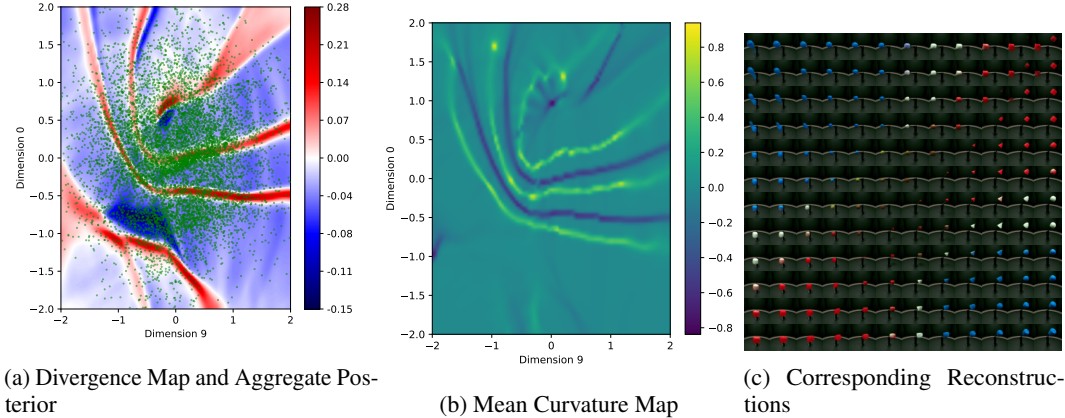

(a) Divergence Map and Aggregate Posterior

(b) Mean Curvature Map

(c) Corresponding Reconstructions

Figure 24: Visualization of the representation learned by the 1-VAE trained on MPI3D Real.

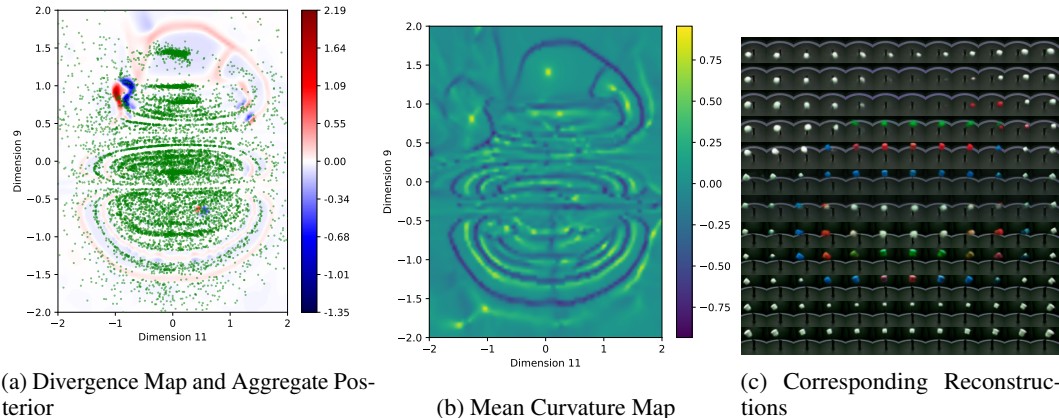

(a) Divergence Map and Aggregate Posterior

(b) Mean Curvature Map

(c) Corresponding Reconstructions

Figure 25: Visualization of the representation learned by the 8-VAE trained on MPI3D Real. Note that due to posterior collapse, the full latent manifold is contained in this projection (see the corresponding response matrix in figure 19).

## B.3 MNIST

Due to the computational cost of evaluating the response function over a dense grid, we focus our visualizations to 2D projections of the latent space. However, for MNIST and Fashion-MNIST, we train several VAE models to embed the whole representation into two dimensions $d = 2$, so that we can visualize the full representation. While the resulting divergence and curvature maps do not demonstrate as intuitive structure as in the disentangled representations for 3D-Shapes or MPI-3D, we can nevertheless appreciate the learned manifold beyond qualitatively observing reconstructions.

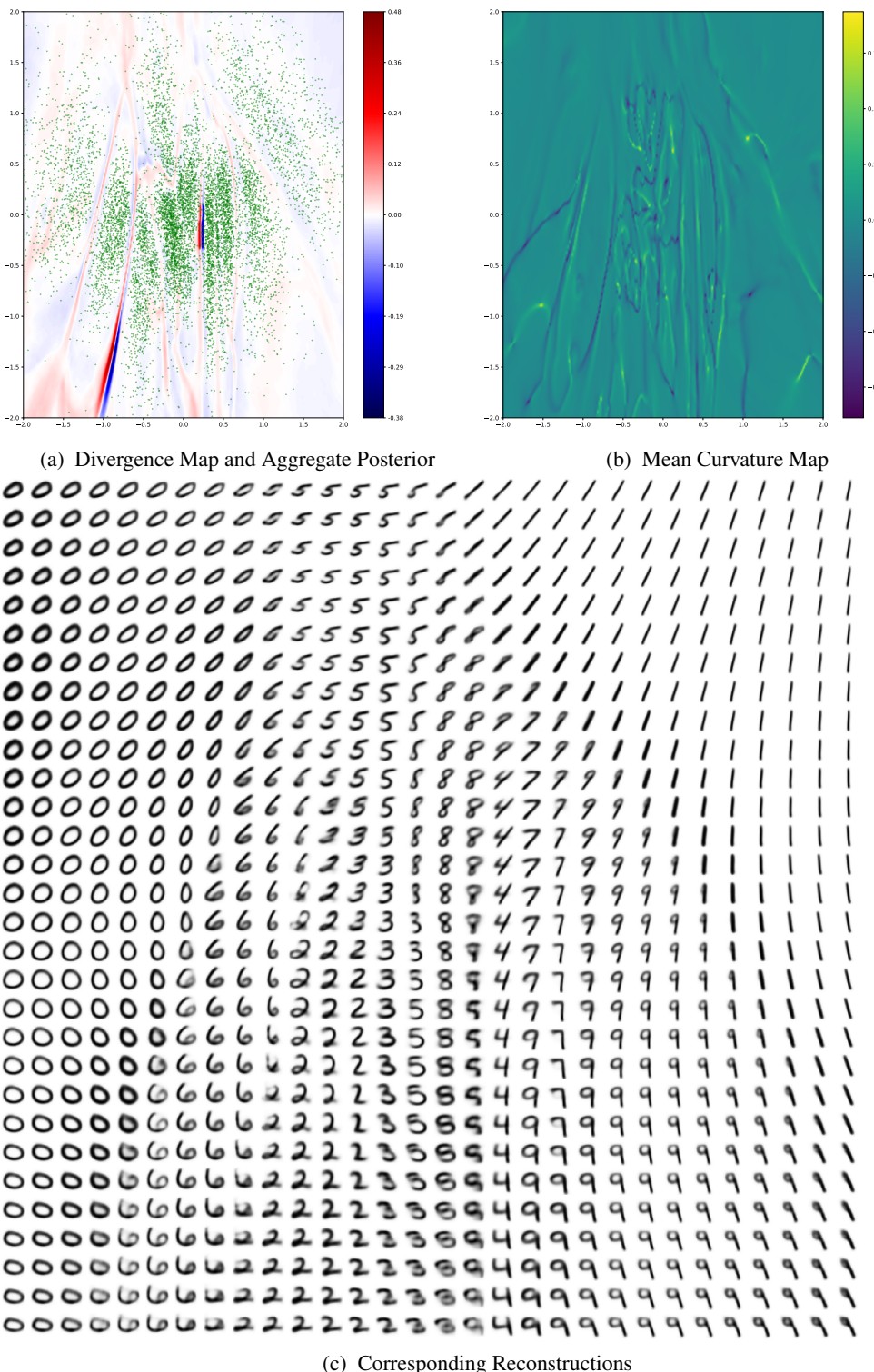

(a) Divergence Map and Aggregate Posterior        (b) Mean Curvature Map

(c) Corresponding Reconstructions

Figure 26: The full latent space for a VAE ($d = 2$) model trained on MNIST. 26a shows the computed divergence of the response field in blue and red while the green points are samples from the aggregate posterior. 26b shows the resulting mean curvature, which identifies 10 points where the curvature spikes and the boundaries between the regions corresponding to different clusters in the posterior. Finally 26c shows the reconstructions over the same region. Note how the high divergence (red) regions correspond to boundaries between significantly different samples (such as changing digit value or stroke thickness).

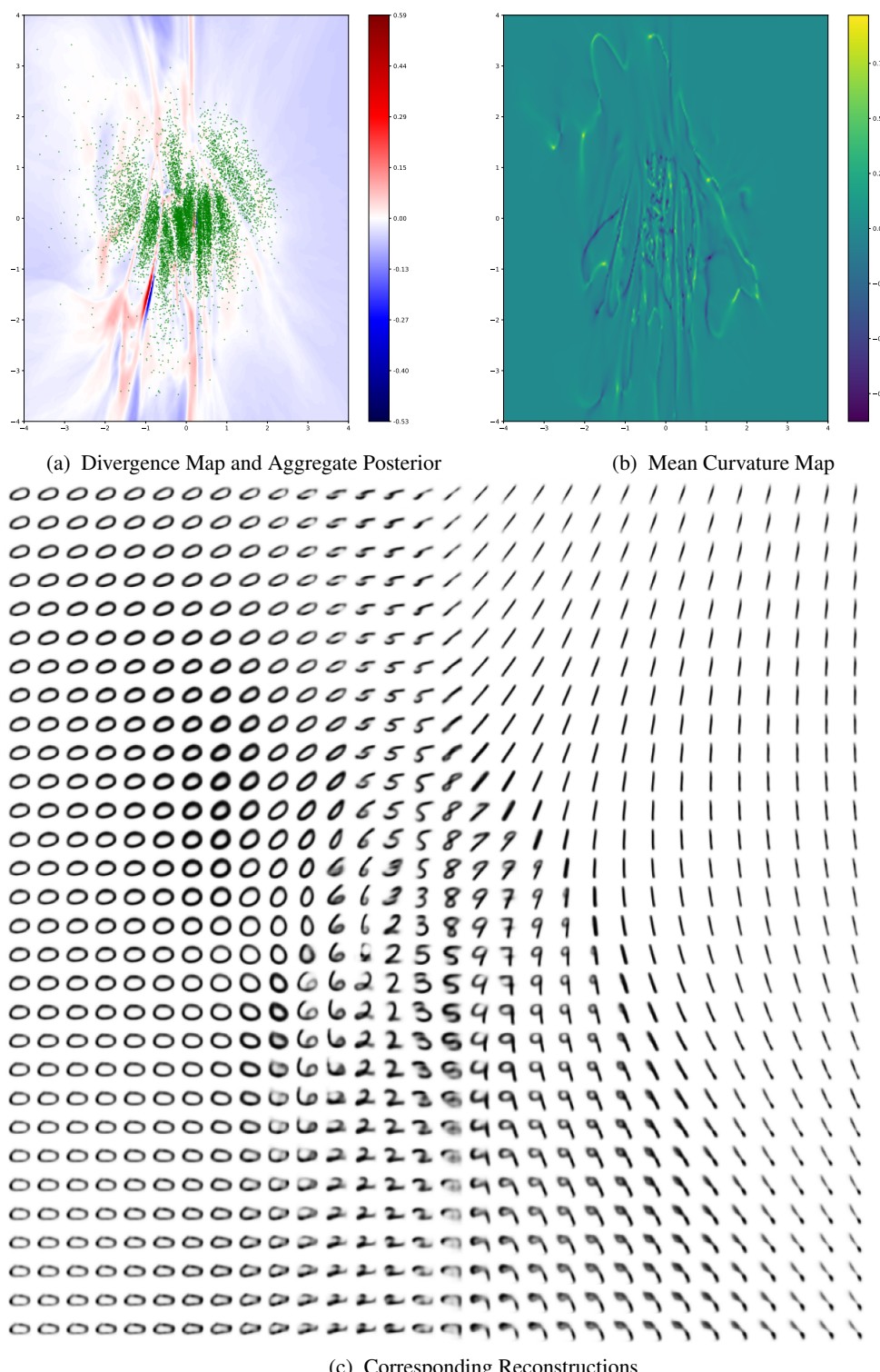

(a) Divergence Map and Aggregate Posterior      (b) Mean Curvature Map

(c) Corresponding Reconstructions

Figure 27: Same plot and model as figure 26, except over a larger range of the latent space $[-4, 4]$. Note that even though the posterior (green dots) is concentrated near the prior (standard normal), reconstructions far away (along the edges of the figure) still look recognizable, demonstrating the exceptional robustness of VAEs to project unexpected latent vectors back onto the learned manifold.

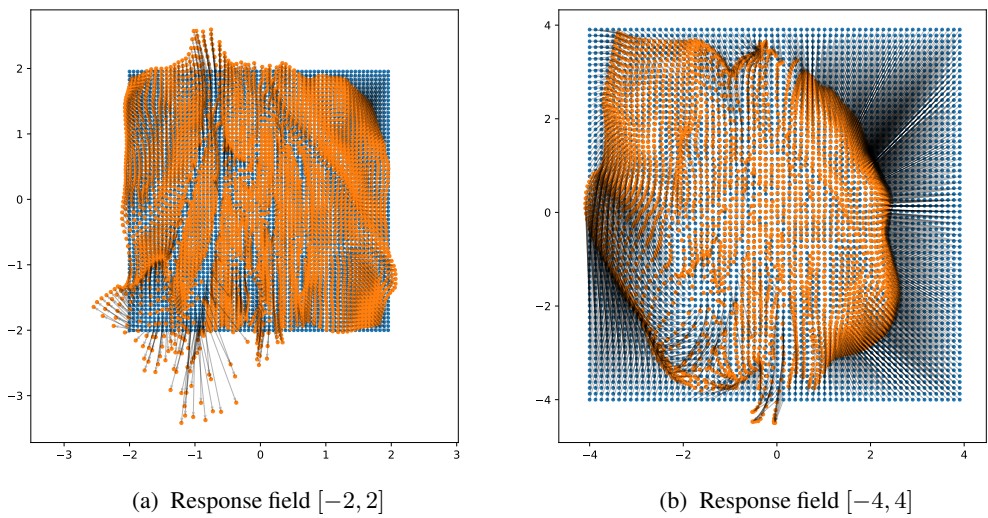

(a) Response field $[-2, 2]$          (b) Response field $[-4, 4]$

Figure 28: Response fields for the same model analyzed in figures 26 and 27. The blue dots show the initial latent samples, and the orange dots connected by the black arrows show the corresponding responses (the latent sample after decoding and reencoding).

## B.3.2 Fashion-MNIST

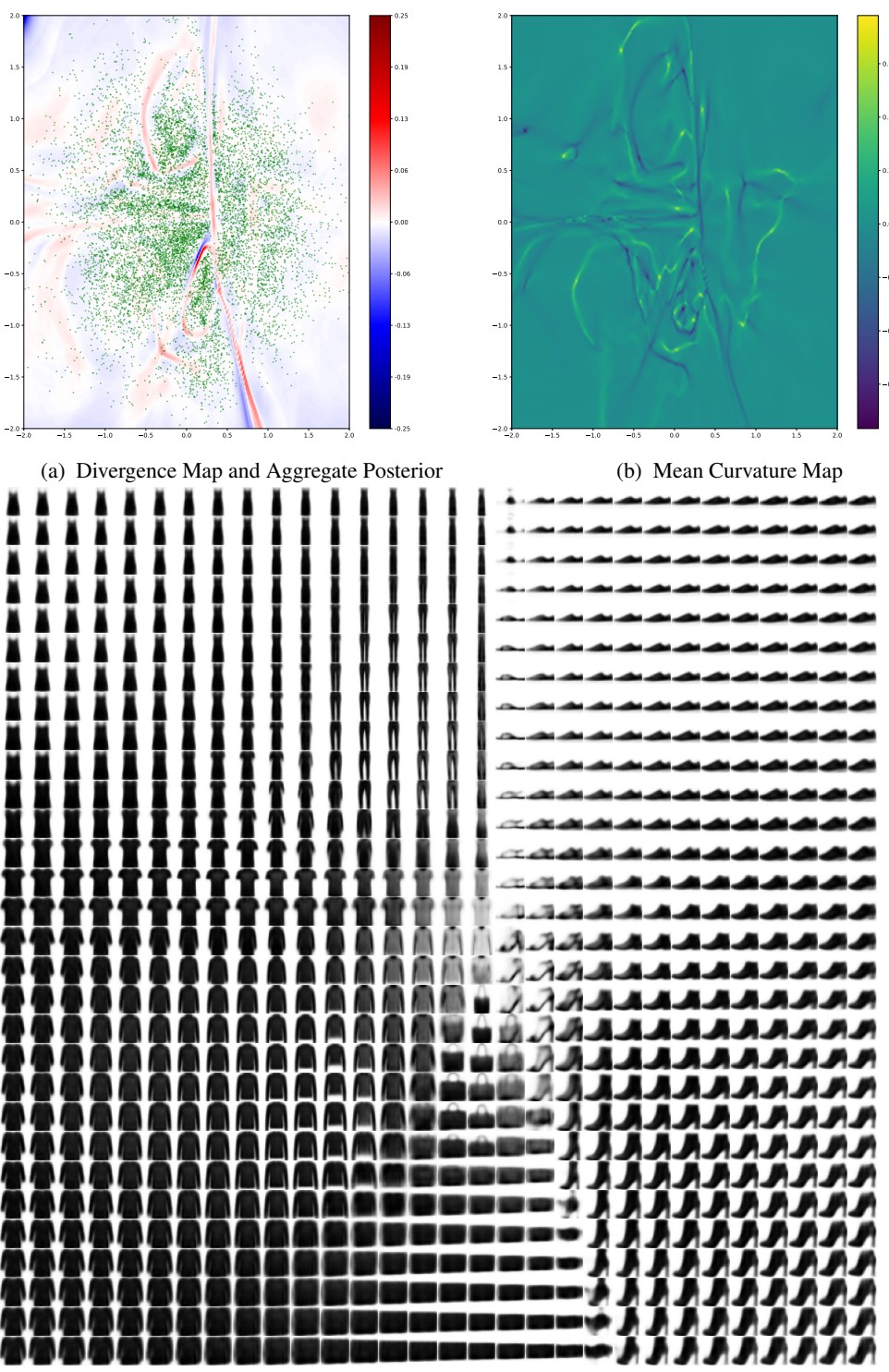

(a) Divergence Map and Aggregate Posterior

(b) Mean Curvature Map

(c) Corresponding Reconstructions

Figure 29: The full latent space for a 8-VAE ($d = 2$) model trained on Fashion-MNIST. 29a shows the computed divergence of the response field in blue and red while the green points are samples from the aggregate posterior. 29b shows the resulting mean curvature, which identifies 10 points where the curvature spikes and the boundaries between the regions corresponding to different clusters in the posterior. Finally 29c shows the reconstructions over the same region. Note how the high divergence (red) regions correspond to boundaries between significantly different samples (such as changing digit value or stroke thickness).

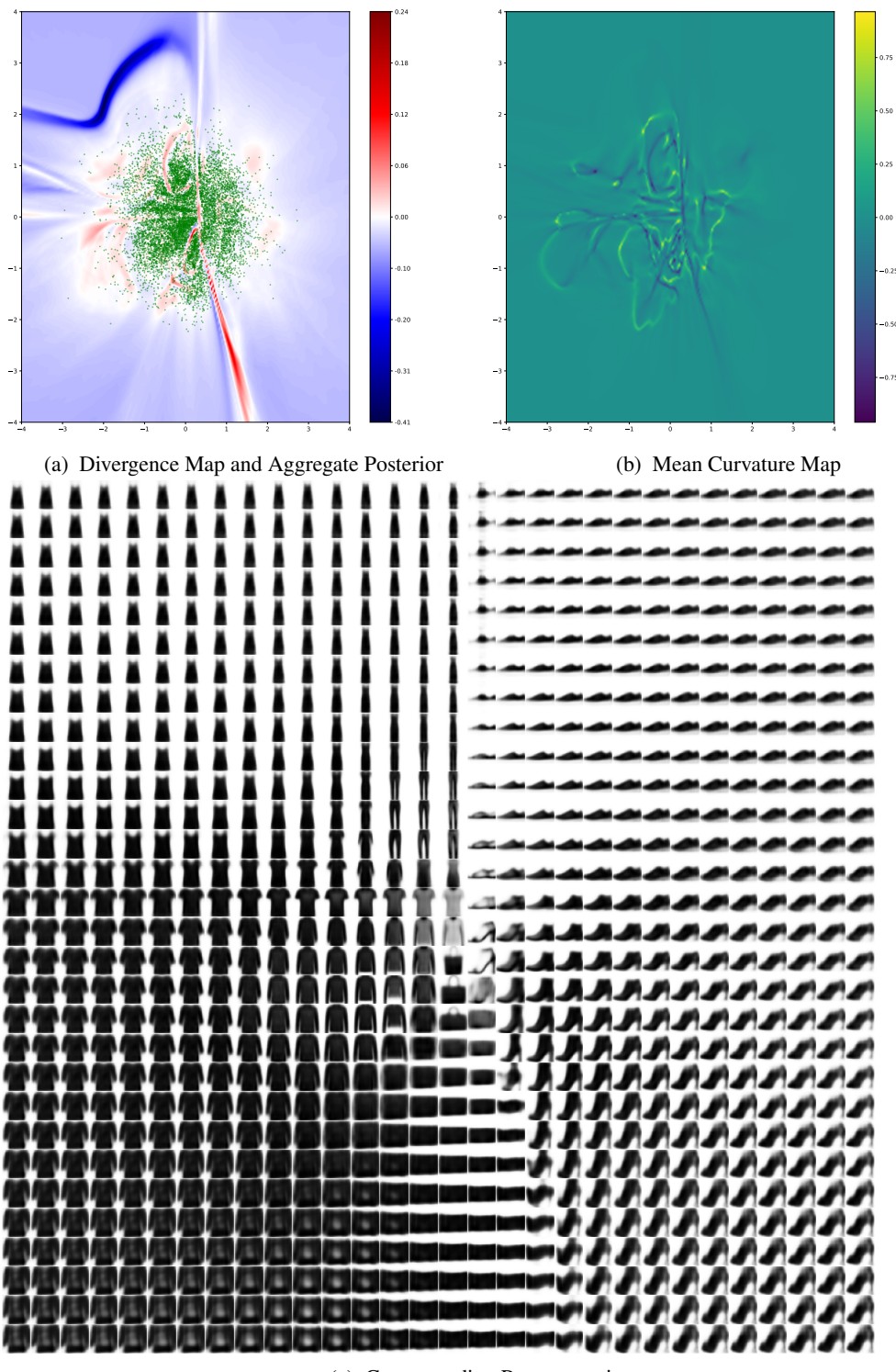

(a) Divergence Map and Aggregate Posterior

(b) Mean Curvature Map

(c) Corresponding Reconstructions

Figure 30: Same plot and model as figure 29, except over a larger range of the latent space $[-4, 4]$. Note that even though the posterior (green dots) is concentrated near the prior (standard normal), reconstructions far away (along the edges of the figure) still look recognizable, demonstrating the exceptional robustness of VAEs to project unexpected latent vectors back onto the learned manifold.

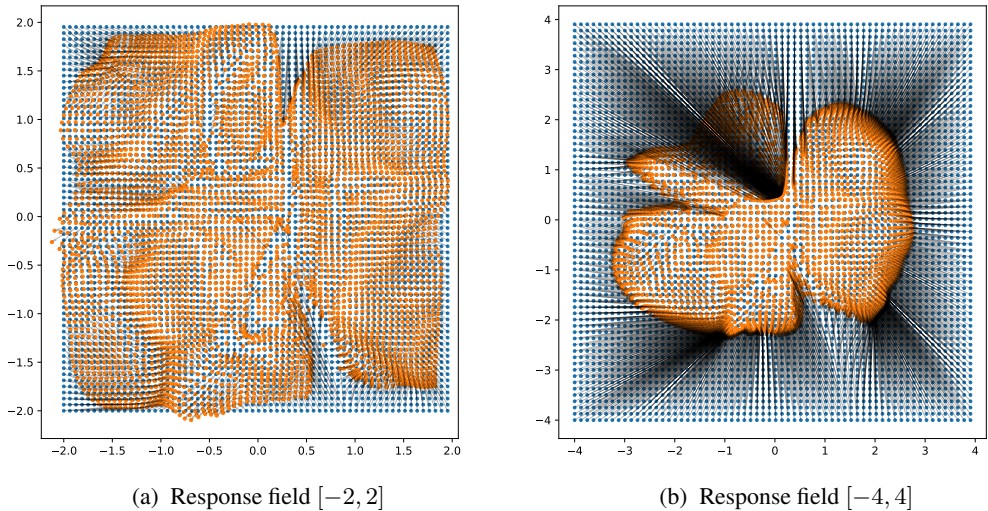

(a) Response field $[-2, 2]$        (b) Response field $[-4, 4]$

Figure 31: Response fields for the same model analyzed in figures 29 and 30. The blue dots show the initial latent samples, and the orange dots connected by the black arrows show the corresponding responses (the latent sample after decoding and reencoding).