# OpenReview forum: "Exploring the Latent Space of Autoencoders with Interventional Assays"
_NeurIPS.cc/2022/Conference — NeurIPS 2022 Accept_

### Official Review · Reviewer_dJu3 · 2022-07-11

**Rating:** 6
**Confidence:** 4
**Soundness:** 3 good
**Presentation:** 3 good
**Contribution:** 3 good

**Summary:**

The paper introduces the latent response framework for evaluating the disentanglement of a representation that is learned by Variational AEs. The framework is based on the so-called latent response function, which takes in input a latent sample, decodes it, and then re-encodes it in order to get a version of the initial latent sample "pushed" into the learned manifold. By performing intervention on individual dimensions of the latent sample, and then applying the latent response function to them, it is possible to quantify how the latent dimensions are causally related to each other and to the true factors of variations (when there are any).


**Questions:**

METHOD:

1. The authors should state in a more explicit way what do they mean for intrinsic and extrinsic curvature of the latent manifold, since they are not standard concepts in the machine learning community.

2. (Line 62) the second term of the ELBO does not encourage the aggregate posterior to match the prior. If that was the case, the term would be  KL(q(Z)|| p(Z)), as q(Z) is the aggregate posterior. The second term it actually encourage the conditional posterior q(Z|X) to match the prior.

3. (Line 132) "the posterior distributions must overlap to, in aggregate, match the prior". As before, the ELBO actually encourages each conditional posterior to match the prior. This sentences can be misleading. You still have the overlapping, as each posterior is pushed towards the standard normal distribution.

4. I would appreciate a complete and explicit derivation of equation 6 to be included in the appendix, since it is one of the main contribution of this work.

5. The method seems to work for standard normal prior, but can it still be used also for other types of priors?

6. The CDS score assumes that a disentangled representation encodes each generative factor into a single separate latent axis. This is, in general a restrictive assumption. Since the latent response matrix can catch the case where a generative factor is connected to more than one latent dimension, I think that the definition of CDS metric should take into account that case.

EXPERIMENTS:

7. The two interpolations of figure 5.c and 5.d do not seem visually differente: I would appreciate to see more examples of latent interpolations on different datasets in the appendix. Also, the caption of figure 5c does not specify which latent dimensions are considered.

8. The matrices of figure 6 show only a subset of latent dimensions. It would seems that, as \beta increases, most of the latent dimensions do not encode any information about the data at all. A deeper discussion on the phenomenon should be added.

9. (lines 305-306) "This may be explained by the DCI-D metric taking additional spurious correlations". The author should provide some evidences of this claim (either via formal proofs or empirically).

10. For the sake of self-containment, the authors should provide a brief definition of responsibility matrix, discussing the differences between it and the conditional response matrix of this work, since judging by Figure 6 the results seems to be quite similar.

MINORS AND TYPOS:
- Line 18: "representations" instead of "representation"
- 32: the reference [31] is cited twice.
- 54: (last word of the line) "are" instead of "and"
- 57: a reference to the reparameterization trick should be included.
- 103-104: "then we could the separation would be trivial"
- 132: "the" instead of "The" after the colon.
- 134: which corresponding.
- 136: missing "." before "Despite".
- 136: "interpretted".
- Overall the sentence from line 132 to line 135 is a bit convoluted and difficult to understand. It should be restated more clearly.
- 138: "from a another" instead of "from an another".
- 191: to measure how quantify how well
- 226: "identifies to how"
- Equation 6: g(s) in the second term does not have the \theta superscript.

**Limitations:**

The final discussion only hints at a (methodological) limitation of their work, that is the requirement for axis-aligned disentanglement. A larger discussion of such a limitation could have been interesting to have.
No discussio of societal impact of the work is provided: maybe the Authors could consider how mixed disentanglement-and-causality methodologies can be supportive of Trustworthy AI.

**Strengths And Weaknesses:**

ORIGINALITY

The paper fits into a quite active and consolidated research topic (studying disentanglement in VAE models) with a twist on more recent and heavily building research area concerning causal learning and causal explanations of deep models. The result is a work which has some value and novelty, contributing both on disentanglement and causality.  What I did not particularly like in the presentation of the work is a certain tendency to underlook relationship with early works on contractive autoencoders and manifold learning in AEs. These are sometimes references in the paper (e.g. [60]) but the relationship is not discussed. Referencing without discussion of the relationship with earlier work is not "useful" for the reader (it only builds up citation counts, which is not the objective of bibliography).

QUALITY and CLARITY

The organization of the content is adequate and so is, mostly, the level of technical depth in the main text. However, there are aspects in the derivation of the main results which can be better clarified (sometimes it suffices to have those in the supplementary materials): please see the Questions section for detailed aspects on the matter. The writing style is not always clear and this is also due to diffused misprints and errors, which could have easily been spotted by a proper proofreading of the work. The authors should spend a little more effort to bring the paper to speed with the expected linguistic and presentation quality of the venue.

SIGNIFICANCE

As stated the key catch underlyign the paper is interesting, though there are some techical questions that needs clarification to convince the reader that everything adds up (again please refer to the Questions section). One open question that is important to address pertains to the generality of the approach: the paper shows how the method can be applied to both vanilla and \beta-VAEs, but it is largely unclear whether the same approach would work with priors different from a single Gaussian.  The experimental setting seems sound, although all the experiments are based on synthetic toy datasets. The experimental results are not always compelling, in particular those on the qualitative side, where it is quite difficult to visually spot the adde value of the proposed approach.

Overall, the work has some merits but there are also open issues that need addressing and clarification as part of the rebuttal discussion. Please refer to questions for such points of discussions and suggestions (also some minors).

---

> ### Author Response · Authors · 2022-08-02
> **Thank you for the detailed feedback**
>
> Thank you for your comprehensive review and detailed feedback.
>
> We appreciate your attention to detail, and we are particularly encouraged that much of your feedback centers around relatively minor alterations, rather than fundamental issues with our method or experiments.
>
> Method
>
> 1. Although we reference a few relevant papers that discuss curvature in the context of manifold learning, we can expand the discussion of past work in the appendix (see A3 and A4).
> 2. You are correct that the ELBO only includes the conditional posterior, not the aggregate posterior. We will clarify that although the loss term is conditioned on a specific sample, since the loss is integrated over all samples, the training converges to matching the posterior to the prior.
> 3. Similar to question 2, we will clarify the misleading statement.
> 4. For completeness, we will include a full derivation in the appendix.
> 5. Although in our experiments and formulation of the method we always use VAEs with a Gaussian prior, all our methods are entirely agnostic to the prior. As long as the prior can easily be sampled from, that is $p(Z)$ and $p(Z_i)$ (which is necessary for generative modeling anyway), new interventions can easily be generated, enabling the computation of the latent response matrix. We chose to focus on a Gaussian prior because it is the most common and offers an illustrative formulation to separate the endogenous $S$ and exogenous $U$ information.
> 6. We agree entirely! Our CDS metric was designed to be a more or less drop-in replacement for commonly used disentanglement score (particularly the DCI metric), all of which focus on the rather restrictive ”single dimension” case. However, the latent response matrix reveals the causal links between latent variables, which can be paired with the conditioned response matrix for a less “naive” notion of disentanglement. While the building blocks are all established, we leave finding a principled approach to connect everything into a structure aware disentanglement metric to future work.
>
> Experiments
>
> 1. The main difference is in the middle images (since that is where the shortest path is maximally far away from the learned manifold), where the sample in 5c shows a blurry shadow on both sides of the wall, while 5d correctly shows no shadow. We will clarify this in the caption.
> 2. As mentioned briefly in the methods section (line 170), the latent response of uninformative latent variables approaches 0 due to posterior collapse (a relatively familiar phenomenon in VAEs), however we will clarify this in the results section.
> 3. In fact, a careful analysis of the model discussed in figure 6 is an example where the DCI score is negatively affected by a spurious correlation in dimension 4.
> 4. We will do so in the appendix. Although the details of the DCI responsibility metric are not crucial to understand our contribution.
>
> Lastly, we can also expand our discussion of the limitations of our methods and include a note on the societal impact. Thanks for the suggestion.

---

> > ### Comment · Reviewer_dJu3 · 2022-08-05
> > **Rebuttal feedback**
> >
> > Many thanks for your careful consideration of the review. I do not have any further pending issue to discuss.

---

### Official Review · Reviewer_4nGX · 2022-07-11

**Rating:** 7
**Confidence:** 3
**Soundness:** 4 excellent
**Presentation:** 4 excellent
**Contribution:** 4 excellent

**Summary:**

This paper concentrates on studying the learnt latent space of VAE. Unlike many existing works which view VAE as simply two parts, reconstruction and alignment between posterior and prior, this paper decomposes the latent variables $Z$ to endogenous and exdogenous components, $S$ and $U$, respectively. This separation of deterministic and stochastic parts of the encoder is indeed quite novel. The paper further proposes a decoding-then-reencoding process to test if the encoder can recognize the resulting sample and if the decoder can filter out exogenous information. This is named as "latent response". Latent responses can help extract the semantic information from the latent space without the need of true label information. Besides, after obtaining the aggregated posterior, each individual dimension's marginal can help build the latent response matrix which reflects the surrounding latent manifold for the input $x$. While this matrix is not informative enough to indicate the relations with semantic meanings, once the label information is available, one can build the conditional response matrix to draw the correspondence. The element scale in the latent response matrix also implies the inter-dimension connections. The resulting analysis tool is applied to a toy example, the double helix, and several common benchmark datasets, including 3D shapes, MNIST, and fashion-MNIST.

**Questions:**

1. Typically, when we say aggregated posterior, it refers to $q(z)=E_x[q(z|x)]$. Could you specify if the subscript $j$ from on page 4? is the index of dimensions? Is there any consideration when you choose square $|\cdot|^2$?
2. On line 181, it is sometimes non-trivial to say if $M_{j_1, j_2}$ approximates $M_{j_2, j_1}$. Do you set a fixed threshold?
3. Why use CDS rather than a more entropy-base metric to evaluate the disentanglement? The current CDS might go better or worse due to some individual $j$'s.
4. How do you determine whether $\hat{s}=s$ for boundaries in response map?
5. If I know the labels, how would you compare your analysis tool with some supervision-guided methods where the separate latent variables are learnt for different semantic variations?
6. This analysis method targets on each individual dims and then re-group some of them. Some other works [1,2] tend to directly learn the factors for different groups. Given that group factor learning is becoming another trending direction, how would you compare your method and these methods?

[1] Li, Y. and Mandt, S., 2018. Disentangled sequential autoencoder. arXiv preprint arXiv:1803.02991.
[2] Bai, J., Wang, W. and Gomes, C.P., 2021. Contrastively disentangled sequential variational autoencoder. Advances in Neural Information Processing Systems, 34, pp.10105-10118.

**Limitations:**

1. It would be nicer if the authors can use the double helix example as the running example to show how to derive the divergence and mean curvature.
2. Since disentanglement and the understanding of the latent space is mostly for the generation tasks, training an additional discriminator to distinguish different factors could be a fun validation experiment.

**Strengths And Weaknesses:**

Pros:
1. This is a very interesting idea. Re-encoding the decoded sample is a novel way to analyze the latent space. Building a correlation matrix for the latent dimensions without any supervision but only interventions is also potentially very helpful for VAE studies.
2. Grouping similar or semantically correlated latent dimensions is always a goal in VAE research. I think this work provides a very useful tool to this end. The proposed analysis tool also facilitates the post-process when some semantic hints become available after a VAE is self-supervisedly trained.

Cons:
1. The latent space is continuous, sometimes it is hard to say if there is any edge between the posterior distributions. For example, when mnist digit "2" transits to "1", it might first become "7". But one cannot say "7" is the boundary.
2. The choice of the square in build the (conditional) response matrix can be elaborated.
3. While the construction of the response matrix $M$ is neat and elegant, I think it should be more clear how the scale in the latent space would affect this matrix. Note that there is no normalized scaling process during VAE encoding. But the euclidean distance can be greatly affected by the scales.
4. It would be great if the analysis tool can be applied to a larger-scale dataset to show its effectiveness.
5. There are many typos and grammar issues in the paper. e.g. "can changes" --> "can change"

---

> ### Author Response · Authors · 2022-08-02
> **Thank you for the insightful questions**
>
> Thank you for your insightful comments and questions. We are particularly encouraged by your positive remarks calling our method “very interesting”, “potentially very helpful”, and “very useful”.
>
> A few short notes on your “cons” section:
>
> 1. Ideally, we would expect to find a boundary (if only a faint one) between the “2” and the “7”, as well as another boundary between the “7” and the “1”. However in the case of handwritten digits, it is not unreasonable to expect a continuous transition from “2” to “1”, in which case there may not be a boundary at all. In practice, this is largely dependent on the amount/quality of data as well as the other inductive biases, since there is no supervision to enforce desired boundaries.
> 2. We aren’t exactly sure what you mean by this: do you mean why some of the variables were removed from the latent response matrix? If so, that is because the others were uninformative due to posterior collapse (see figure 12 for the full matrix). In any case, we will be happy to address further questions during the discussion period.
> 3. The prior dictates the scale, so as long as the scale of the prior is constant across all latent dimensions (such as in a standard normal), the responses between dimensions can be compared directly.
>
> And to answer your questions:
>
> 1. Yes, the subscript $j$ refers to the $j$th latent variable (see line 165). The distance metric was chosen such that the latent response matrix computes the root-mean-squared effect of an intervention, but there is no particular reason to prefer this metric over another.
> 2. Correct, in this case, we manually inspected the matrices to see if any pairs were notably similar, which prompted deeper analysis. However, in general, as you point out, it may be challenging to determine which latent dimensions should be grouped, unless the representation is already somewhat disentangled.
> 3. The CDS is based on the entropy up to a constant offset due to conditioned response matrix not being normalized (this mirrors the DCI disentanglement score computation from the responsibility matrix, see [1] for details). If you interpret the elements of the conditioned response matrix as probabilities of a given latent variable affecting a given true factor, then computing the entropy follows analogously as in the DCI disentanglement score, however, since the conditioned response matrices can also be interpreted as the average effect an intervention on a latent variable has on a true factor, the statistical interpretation is somewhat of an oversimplification. To avoid these ambiguities, we leave the matrix as is, which, provided the scale of the prior is constant, does not bias the score towards any true factor.
>
>     [1] Cian Eastwood and Christopher KI Williams. A framework for the quantitative evaluation of disentangled representations. 2018.
>
> 4. For the response maps we start with a collection of latent samples z with unknown s, so the $\hat{s}$ is treated as the s, from which the spatial divergence of the noise $u(z) = \hat{s} - z$, is used for identifying the boundaries. When the divergence is positive (divergent) that corresponds to a boundary.
> 5. With supervision, the latent responses can still be used to identify the causal relationships between latent variables learned by the model. In fact, since full supervision would make it much easier to interpret the latent variables, the learned causal links could be used to evaluate how different inductive biases affect the causal structure of the resulting learned generative model.
> 6. In principle, the latent response framework readily extends to multi-dimensional latent variables (in fact, as we discuss in the paper, the latent response matrix can actually be used to group individual latent variables based on the information they encode). The main limitation is in quantifying the similarity between different values of a latent variable, which becomes more challenging for multi-dimensional variables since there may be non-trivial structure beyond dimensions within the same variable.
>
>     Consequently, you can think of this as a “bottom-up” approach to finding links between latent dimensions to group them together, while a more common approach (such as in the papers you reference and in other approaches that structure the representation using self-supervised losses) is the “top-down” where the model architecture or loss terms group together latent dimensions during training (such as style and content or dynamic and static features).

---

### Official Review · Reviewer_g1w9 · 2022-07-15

**Rating:** 7
**Confidence:** 2
**Soundness:** 3 good
**Presentation:** 2 fair
**Contribution:** 4 excellent

**Summary:**

This paper describes an approach to understanding the structure in VAE models. At the core of this approach is what the authors term the latent response map, the function obtained by first decoding and the re-encoding a point in the latent space. This is used to build a latent response matrix, which measures the degree to which modifying each latent variable affects the values of each other latent variable. This can then be used to understand the underlying causal structure in the latent space underpinning a VAE.

In cases where the true generative model is known an extension of this, which the authors term the conditioned response matrix, follows a similar approach to measure the degree to which changes in a true generating factor influences each latent factor. This can then be used to derive a new disentanglement metric.

Finally, the latent response function can also be used to build what the authors term a response map, which uses the latent response function to estimate the mean curvature in the latent manifold. This provides insight into the structure of the latent space and can be used to efficiently compute approximate geodesics between latent samples (if I understand correctly)

The method is first evaluated on simple toy data: a 3D double-helix generated from 2D latent space (samples from a uniform and Bernoulli r.v.). This is used to illustrate how the method gives insight into the learned latent manifold and can be used to meaningfully interpolate between points in the latent space

The method is then evaluated on the 3D-shapes dataset, a synthetic data set that is often used for benchmarking methods for learning disentangled representations. This is used to demonstrate:
(i) the mean curvature map captures meaningful structure in the latent space - the clusters implied by the curvature map for a pair of latent variables coincide with different floor colours in the generated data;
(ii) the response map can be used to meaningfully interpolate between points in the latent space; and
(iii) the disentanglement score inferred from the conditioned response matrix aligns well with other metrics of disentanglement




**Questions:**

None

**Strengths And Weaknesses:**

Strengths

- The method described in this paper seems novel and tackles important problems in representation learning

Weaknesses

- I found the writing in the paper difficult to follow at times. For example, in the explanation of the conditioned latent response matrix it took me a while to understand "subset of observations which are all fully identical except for the factor Yc" referred to a subset of observations which agree on all other true factors are fixed and which take different values for Yc. I initially thought "fully identical" referred to agreeing on X also.

- Empirical evaluation was limited to toy / synthetic data. For a paper like this, I feel this is just about sufficient but the paper would certainly be strengthened by the application of the method to real data

---

> ### Author Response · Authors · 2022-08-02
> **Thank you for the thoughtful comments**
>
> Thank you for the time you spent and your thoughtful comments. We appreciate the fact that on balance you gave us a positive score (“accept”) after taking into account all contributions (conceptual novelty and experiments). We agree that the paper can be further strengthened and we will clarify the point you are referring to. For us, this is the starting point of a line of work and we would certainly work on further applications, yet we feel (and hope you agree) that this is mature enough to be exposed to the community at this point.

---

### Author Response · Authors · 2022-08-02
**General response about the paper revision posted**

Overall, we are encouraged by the largely positive responses from the reviewers, all of which expressed an interest in our project, and thank everyone for their feedback and support.

Based on all the reviewers’ helpful feedback, we have made numerous minor changes to the paper including adding several clarifying sections to the appendix. Additionally, we have tried to fix all the typos in the new version and will have a native speaker check it in detail before the final version.

We hope this revision will further help the reviewers’ confidence to reach a consensus on the value of our work and foster further discussion.

---

### Meta-Review · Area_Chair_CeCa · 2022-08-27

**Recommendation:** Accept
**Confidence:** Certain

**Metareview:**

This paper proposes a method to analyze the latent space of autoencoders, by decoding a point in the latent space and then encoding it back, which can be used to build response maps that measure how much a latent dimension changes when some other dimension changes.  All reviewers liked the idea, and I support accepting the paper.  The paper still contains some presentation issues, the authors should make an effort to improve the readability of the paper and make it more accessible following the suggestions of the reviewers.

**Award:**

No

---

### Decision · Program_Chairs · 2022-09-14

Accept